# Differentiable Synthesis of Program Architectures

**Guofeng Cui**
Department of Computer Science
Rutgers University
gc669@cs.rutgers.edu

**He Zhu**
Department of Computer Science
Rutgers University
hz375@cs.rutgers.edu

## Abstract

Differentiable programs have recently attracted much interest due to their interpretability, compositionality, and their efficiency to leverage differentiable training. However, synthesizing differentiable programs requires optimizing over a combinatorial, rapidly exploded space of program architectures. Despite the development of effective pruning heuristics, previous works essentially enumerate the discrete search space of program architectures, which is inefficient. We propose to encode program architecture search as learning the probability distribution over all possible program derivations induced by a context-free grammar. This allows the search algorithm to efficiently prune away unlikely program derivations to synthesize optimal program architectures. To this end, an efficient gradient-descent based method is developed to conduct program architecture search in a continuous relaxation of the discrete space of grammar rules. Experiment results on four sequence classification tasks demonstrate that our program synthesizer excels in discovering program architectures that lead to differentiable programs with higher $F_1$ scores, while being more efficient than state-of-the-art program synthesis methods.

## 1 Introduction

Program synthesis has recently emerged as an effective approach to address tasks in several fields where deep learning is applied traditionally. A synthesized program in a domain-specific language (DSL) provides a powerful abstraction for summarizing discovered knowledge from data and offers greater interpretability and transferability across tasks than a deep neural network model, while achieving competitive task performance [1–4].

A differentiable program encourages interpretability by using structured symbolic primitives to compose a set of differentiable modules with trainable parameters in its architecture. These parameters can be efficiently learned with respect to a differentiable loss function over the program's outputs. However, synthesizing a reasonable program architecture remains challenging because the architecture search space is discrete and combinatorial. Various enumeration strategies have been developed to explore the program architecture space, including greedy enumeration [1, 2], evolutionary search [5], and Monte Carlo sampling [6]. To prioritize highly likely top-down search directions in the combinatorial architecture space, NEAR [7] uses neural networks to approximate missing expressions in a partial program whose $F_1$ score serves as an admissible heuristic to effective graph search algorithms such as A* [8]. However, since the discrete program architecture search space is intractably large, enumeration-based search strategies are inefficient in general.

We propose to encode program architecture search as learning the probability distribution over all possible program architecture derivations induced by a context-free DSL grammar. This problem bears similarities with searching the structure of graphical models [9] and neural architecture search. Specifically, to support differentiable search, DARTS [10] uses a composition of softmaxes over all possible candidate operations between a fixed set of neural network nodes to relax the discrete search space of neural architectures. However, applying this method to program synthesis is challenging

35th Conference on Neural Information Processing Systems (NeurIPS 2021).

because the program architecture search space is much richer [7]. Firstly, different sets of operations take different input and output types and may only be available at different points of a program. Secondly, there is no fixed bound on the number of expressions in a program architecture.

To address the aforementioned challenges, we learn the probability distribution of program architectures in a continuous relaxation of the search space of DSL grammar rules. We conduct program architecture search in a *program derivation graph*, in which nodes encode architectures with missing expressions, and paths encode top-down program derivations. For each partial architecture $f$ on a graph node, we relax the categorical choice of production rules for expanding a missing expression in $f$ to a softmax over all possible production rules with trainable weights. A program derivation graph essentially expresses all possible program derivations under a context-free grammar up to a certain depth bound (on the height of program abstract syntax trees), which can be iteratively increased during search to balance accuracy and architecture complexity. We encode a program derivation graph itself as a differentiable program whose output is weighted by the outputs of all the programs involved. We seek to optimize program architecture weights with respect to an accuracy loss function defined over the encoded program's output. The learned weights allow our synthesis algorithm to efficiently prune away search directions to unlikely program derivations to discover optimal programs. Compared with enumeration-based synthesis strategies, differentiable program synthesis in the relaxed architecture space is easier and more efficient with gradient-based optimization.

One major challenge of differentiable program architecture synthesis is that a program derivation graph involves an exponential number of programs and a huge set of trainable variables including architecture weights and program parameters. To curb the large program derivation search space, we introduce node sharing in program derivation graphs and iterative graph unfolding. Node sharing allows two partial architectures to share the same child nodes if the missing expressions in the two architectures can be expanded using the same grammar rules. Iterative graph unfolding allows the synthesis algorithm to construct a program derivation graph on the fly focusing on higher-quality program derivations than all the rest. These optimization strategies significantly reduce the program architecture search space, scaling differentiable program synthesis to real-world classification tasks. We evaluate our synthesis algorithm in the context of learning classifiers in sequence classification applications. We demonstrate that our algorithm substantially outperforms state-of-the-art methods for differentiable program synthesis, and can learn programmatic classifiers that are highly interpretable and are comparable to neural network models in terms of accuracy and $F_1$-scores.

As a summary, this paper makes three contributions. Firstly, we encode program synthesis as learning the probability distribution of program architectures in a continuous relaxation of the discrete space defined by programming language grammar rules, enabling differentiable program architecture search. Secondly, we instantiate differentiable program architecture synthesis with effective optimization strategies including node sharing and iterative graph unfolding, scaling it to real-world classification tasks. Lastly, we present state-of-the-art results in learning programmatic classifiers for four sequence classification applications.

## 2  Problem Formulation

A program in a domain-specific language (DSL) is a pair $(\alpha, \theta)$, where $\alpha$ is a discrete program architecture and $\theta$ is a vector of real-valued parameters of the program. Given a specification over the intended input-output behavior of an unknown program, program synthesis aims to discover the program's architecture $\alpha$ and optimize the program parameters $\theta$.

In this paper, we focus on learning programmatic classifiers for sequence classification tasks [11]. We note that the proposed synthesis technique is applicable to learning any differentiable programs.

**Program Architecture Synthesis.** A program architecture $\alpha$ is typically synthesized based on a context-free grammar [12]. Such a grammar consists of a set of production rules $\alpha_k \rightarrow \{\sigma_j\}_{j=0}^{J}$ over terminal symbols $\Sigma$ and nonterminal symbols $Y$ where $\alpha_k \in Y$ and $\sigma_j \in \Sigma \cup Y$. As an example, consider the context-free grammar of a DSL for sequence classification depicted in the standard Backus-Naur form [13] in Fig. 1, adapted from [7]. A terminal in this grammar is a symbol that can appear in a program's code, e.g. $x$ and the **map** function symbol, while a nonterminal stands for a missing expression (or subexpression), e.g. $\alpha_2$ and $\alpha_3$. Any program in the DSL operates over a real vector or a sequence of real vector $x$. It may use constants $c$, arithmetic operations **Add** and **Multiply**, and an **I**f-**T**hen-**E**lse branching construct **ITE**. To avoid discontinuities for

$$\alpha ::= x \mid c \mid \textbf{Add} \ \alpha_1 \ \alpha_2 \mid \textbf{Multiply} \ \alpha_1 \ \alpha_2 \mid \textbf{ITE} \ \alpha_1 \geq 0 \ \alpha_2 \ \alpha_3 \mid \textbf{F}_{\textbf{S},\theta}(x) \mid \textbf{map} \ (\textbf{fun} \ x_1.\alpha_1) \ x \mid$$
$$\textbf{mapprefix} \ (\textbf{fun} \ x_1.\alpha_1) \ x \mid \textbf{fold} \ (\textbf{fun} \ x_1.\alpha_1) \ c \ x \mid \textbf{SlideWindowAvg} \ (\textbf{fun} \ x_1.\alpha_1) \ x$$

Figure 1: Context-free DSL Grammar for Sequence Classification (adapted from [7]).

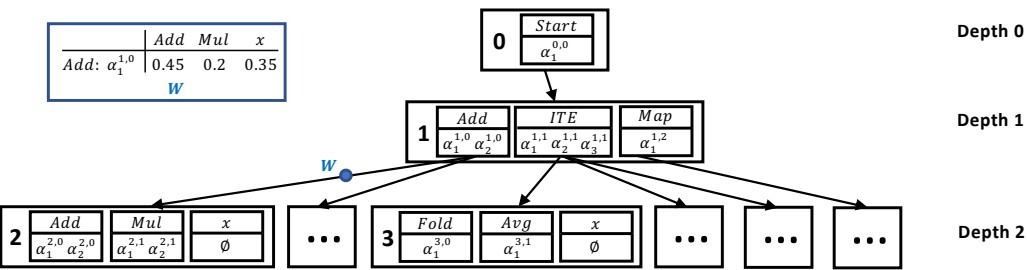

Figure 2: Program Derivation Graph of the grammar in Fig. 1.

differentiability, we interpret it in terms of a smooth approximation: $[\![\textbf{ITE}(\alpha_1 \geq 0, \alpha_2, \alpha_3)]\!](x) = \sigma([\![\alpha_1]\!](x)) \cdot [\![\alpha_2]\!](x) + (1 - \sigma([\![\alpha_1]\!](x)) \cdot [\![\alpha_3]\!](x)$ where $\sigma$ is the sigmoid function. A program may invoke a customized library of differentiable, parameterized functions. In our context, these functions are in the shape $\textbf{F}_{S,\theta}(x)$ that extract a vector consisting of a predefined subset $S$ of the dimensions of an input $x$ and pass the extracted vector through a linear function with trainable parameters $\theta$. A program may also use a set of higher-order combinators to recurse over sequences including the standard **map** and **fold** combinators. The higher-order combinators takes as input an anonymous function **fun** $x.e(x)$ that evaluates an expression $e(x)$ over the input $x$. For a sequence $x$, the **mapprefix** higher-order combinator returns a sequence $f(x[1:1]), f(x[1:2]), \ldots, f(x[1:n])$, where $x[1:i]$ is the $i$-th prefix of $x$. The **SlideWindowAvg** function computes the average of a sequence over a moving window of frames.

We define the complexity of a program architecture $\alpha$. Let each grammar rule $r$ have a non-negative real-valued cost $c(r)$. The structural cost $c(\alpha)$ is the sum of the costs of the multi-set of rules used to create $\alpha$. Intuitively, program architectures with lower structural cost are more interpretable. In the context of this paper, program synthesis aims to learn a simple program (in terms of structure cost) that satisfies some specifications over program input-output behaviors. In this paper, we set $c(r) = 1$ for any production rule $r$.

**Program Synthesis Specifications.** In a sequence classification task, a set of feature sequences $\{i_k\}_{k=1}^{K}$ are taken as input and we expect to classify $i_k$ into a certain category $o_k$. Each $i_k$ is a sequence of observations. Each observation captures features extracted at a frame as a 1-dimensional real-valued vector. We aim to synthesize a program $P(\cdot; \ \alpha, \theta)$ as a classifier with high accuracy and low architecture cost. Our program synthesis goal is formalized as follows:

$$\underset{\theta, \alpha}{\arg \min} \ \mathbb{E}_{i_k, o_k \sim D}[\ell(P(i_k; \ \alpha, \theta), o_k)] + c(\alpha) \tag{1}$$

where $D(i_k, o_k)$ is an unknown distribution over input sequences $i_k$ and labels $o_k$. The first term of Equation (1) defines some prediction error loss $\ell$ of a program $P(\cdot)$ for a classification task over $P$'s predicted labels and the ground truth labels. The second term enforces program synthesis to learn an architecturally simple programmatic classifier.

## 3 Differentiable Program Architecture Synthesis

We formulate program architecture derivation as a form of top-down graph traversal. Given the context-free grammar $\mathcal{G}$ of a DSL, an architecture derivation starts with the initial nonterminal (i.e. the empty architecture), then applies the production rules in $\mathcal{G}$ to produce a series of partial architectures which consist in expressions made from one or more nonterminals and zero or more terminals, and terminates when a complete architecture that does not include any nonterminals is derived.

Formally, program architecture synthesis with respect to a context-free grammar $\mathcal{G}$ is performed over a directed acyclic *program derivation graph* $G = \{V, E\}$ where $V$ and $E$ indicate graph nodes and edges. Fig. 2 depicts a program derivation graph for the sequence classification grammar in Fig. 1. A node $u \in V$ is a set of partial or complete program architectures permissible by $\mathcal{G}$. An edge $(u, u') \in E$ exists if one can obtain the architectures in $u'$ by expanding a nonterminal of an architecture in $u$ following some production rules of $\mathcal{G}$. For simplicity, Fig. 2 only shows three partial or complete architectures in any node of the program derivation graph. In the graph node at depth 1, we expand the initial nonterminal $\alpha_1^{0,0}$ to the **Add**, **ITE** and **Map** functions (each with missing expressions) using the grammar rules in Fig. 1. Notice that the edge direction in a program derivation graph indicates search order. However, program dataflow through each edge $(u, u')$ is in the opposite direction. The output of $u'$ is calculated first and then passed as input to $u$.

The main challenge of program architecture synthesis is that the search space embedded in a program derivation graph is discrete and combinatorial. Enumeration-based synthesis strategies are inefficient in general because of the intractable search space. Instead, we aim to learn the probability distribution of program architectures within a program derivation graph in a continuous relaxation of the search space. Specifically, to expand a nonterminal of a partial program architecture, we relax the categorical choice of production rules in a context-free grammar into a softmax over all possible production rules with trainable weights. For example, in Fig. 2, if we expand the initial nonterminal $\alpha_1^{0,0}$ to a partial architecture **Add** $\alpha_1^{1,0}$ $\alpha_2^{1,0}$ on node 1, we have several choices to further expand the architecture's first nonterminal $\alpha_1^{1,0}$, weighted by the probability matrix $w$ (obtained after softmax) drawn in Fig. 2. Based on $w$, the synthesizer will choose to expand $\alpha_1^{1,0}$ to **Add** $\alpha_1^{2,0}$ $\alpha_2^{2,0}$ on node 2. Our main idea to learn architecture weights is to encode a program derivation graph itself as a differentiable program $\mathcal{T}_{w,\theta}$ whose output is weighted by the outputs of all programs included in $\mathcal{T}_{w,\theta}$, where $w$ represents architecture weights and $\theta$ includes program parameters of all the mixed programs in the graph. The parameters $w$ and $\theta$ can be jointly optimized with respect to a differentiable loss function $\ell$ over program outputs via bi-level optimization. Similar to DARTS [10], we train $\theta$ and $w$ on a parameter training dataset $D_\theta$ and an architecture validation dataset $D_w$ respectively until convergence:

$$\theta' = \theta - \nabla_\theta \mathbb{E}_{i_k,o_k \sim D_\theta} \ell\big(\mathcal{T}_{w,\theta}(i_k), o_k\big)$$
$$w' = w - \nabla_w \mathbb{E}_{i_k,o_k \sim D_w} \ell\big(\mathcal{T}_{w,\theta'}(i_k), o_k\big)$$

(2)

However, a program derivation graph includes an exponential number of programs. Therefore, it involves a huge set of trainable variables including program architecture weights $w$ and unknown program parameters $\theta$. To curb the large program derivation search space, we introduce node sharing (Sec. 3.1) and iterative graph unfolding (Sec. 3.2).

## 3.1 Node Sharing

Intuitively, node sharing in a program derivation graph allows two partial architectures to share the same child nodes if the nonterminals in the two architectures can be expanded using the same grammar production rules. Fig. 3 depicts the compressed program derivation graph for the sequence classification grammar in Fig. 1. At depth 1, three partial architectures **Add** $\alpha_1^{1,0}$ $\alpha_2^{1,0}$, **ITE** $\alpha_1^{1,1} \geq 0$ $\alpha_2^{1,1}$ $\alpha_3^{1,1}$, and **Map** (**fun** $x_1.\alpha_1^{1,2}$) are expanded from the initial nonterminal $\alpha_1^{0,0}$. Because only one of the three partial architectures would be used to derive the final synthesized program, we allow the nonterminals $\alpha_2^{1,0}$, the second parameter of **Add**, and $\alpha_2^{1,1}$, the second parameters of **ITE**, to share the same child node 3, weighted by the probability matrix $w$ drawn in Fig. 3. Importantly, node sharing takes function arities and types into account. The matrix $w$ has 0 probability for the **Map** partial architecture because unlike **Add** and **ITE** it does not contain a second parameter.

Formally, in a program derivation graph, let $K_u$ be the number of program architectures on node $u$. Denote $f_k^u\big(\alpha_1^{u,k}, \ldots, \alpha_{\eta(f_k^u)}^{u,k}\big)$ as the $k$-th (partial) architecture on $u$ where $\eta(f_k^u)$ is the number of nonterminals contained in $f_k^u$ and $\alpha_i^{u,k}$ is the $i$-th nonterminal of $f_k^u$. For the grammar of Fig. 1, essentially each $f_k^u$ is a function application with missing argument expressions $\alpha_i^{u,k}$, $1 \leq i \leq \eta(f_k^u)$, and $\eta(f_k^u)$ is the arity of the function. Assume that $u'$ is the $i$-th child of $u$ from left to right in the program derivation graph. The weight $w_e$ of the edge $e = (u, u')$ is of the shape $\mathbb{R}^{K_u \times K_{u'}}$ where the matrix rows refer to the partial architectures on $u$ and the matrix columns refer to architectures on $u'$. We have $w_e[(k, k')]$ proportional to the probability of expanding the $i$-th nonterminal of $f_k^u$ to $f_{k'}^{u'}$

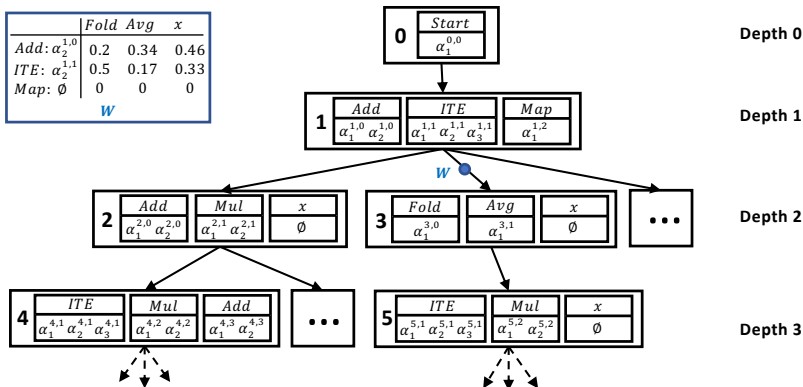

Figure 3: Node Sharing on Program Derivation Graphs.

(obtained after softmax), e.g. the $w$ matrix drawn in Fig. 3. To make the architecture search space continuous, we relax the categorical choice of expanding a particular nonterminal $\alpha_i^{u,k}$ to a softmax over all possible grammar production rules for $\alpha_i^{u,k}$ in the program derivation graph:

$$[\![\alpha_i^{u,k}]\!](x) = \sum_{k'=0}^{K_{u'}} \frac{\exp(w_e[(k,k')])}{\sum_{j=0}^{K_{u'}} \exp(w_e[(k,j)])} \cdot [\![f_{k'}^{u'}\big(\alpha_1^{u',k'}, \ldots, \alpha_{\eta(f_{k'}^{u'})}^{u',k'}\big)]\!](x)$$ (3)

$$\text{where } u' \text{ is the } i\text{-th child of } u \text{ and } e = (u, u')$$

**Complexity**. Let $D$ be the depth of a program derivation graph, $K_{max}$ be the number of productions rules, and $\eta_{max}$ be the maximum number of nonterminals in any rules of a context-free grammar. With node sharing, we reduce the space complexity of the program derivation graph from $O([K_{max} \cdot \eta_{max}]^{D+1})$ to $O([\eta_{max}]^{D+1})$. In a program synthesis task, $K_{max}$ is typically much larger than $\eta_{max}$. Without compression, the program derivation graph of a grammar with a large $K_{max}$ even hardly fits GPU memory.

### 3.2 Iterative Graph Unfolding

Node sharing significantly restricts the width of a program derivation graph. However, a derivation graph still grows exponentially with its depth, which limits the scalability of differentiable architecture search. To address this problem, we propose an on-the-fly approach that unfolds program derivation graphs iteratively and prunes away unlikely candidate architectures at the end of each iteration based on their weights. Fig. 4 depicts the iterative procedure of derivation graph unfolding.

At the initial iteration, the program derivation graph is shallow as it only contains architectures up to depth $d_s$. We set $d_s = 2$ in Fig. 4. For any partial program architecture $f_k^u\big(\alpha_1^{u,k}, \ldots, \alpha_{\eta(f_k^u)}^{u,k}\big)$ on any node $u$ of the depth-bounded graph, our algorithm substitutes neural networks for the nonterminals $\alpha_i^{u,k}$ to approximate the missing expressions. These networks are type-consistent. For example, a recurrent neural network is used to replace a missing expression whose inputs are supposed to be sequences. For a program derivation graph such, the unknown program parameters $\theta$ come from both parameterized functions like $f_k^u$ and the neural modules. The goal is to optimize the architecture weights and the unknown program parameters using Equation 2. The qualities of candidate partial architectures on each graph node are ranked based on the learned architecture weights.

In the next iteration, on each graph node, our synthesis algorithm retains top-$N$ program architectures as children for each partial architecture on the node's parent, which are defined to be those assigned with higher weights on the node's incoming edge in the previous iteration. We set $N = 2$ in the example of Fig. 4. After top-$N$ preservation on each node, our synthesis algorithm increases the depth of the program derivation graph by expanding the nonterminals (that were replaced with neural modules in the previous iteration) $d_s$ depths deeper. Suitable neural networks are leveraged to substitute any new nonterminals at depth $2d_s + 1$. Our algorithm again jointly optimizes architecture weights and unknown program parameters and performs top-$N$ preservation on each node based on

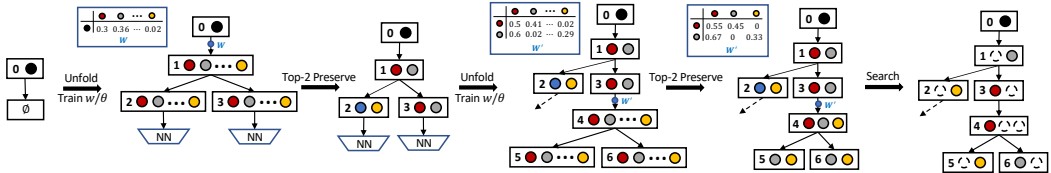

Figure 4: Differentiable program architecture synthesis with iterative graph unfolding.

learned architecture weights, as depicted in Fig. 4. Such a process iterates until the unfolded program derivation graph contains no nonterminals or the maximum search depth is reached. Our differentiable program architecture synthesis method is outlined in (the first while loop of) Algorithm 1.

### 3.3 Searching Optimal Programs

Once we have an optimized program derivation graph $G$, due to the top-$N$ preservation strategy, each node retains a small number of partial architectures. From $G$, we could greedily obtain a discrete program architecture top-down by replacing each graph node containing mixed partial architectures with the most likely partial architecture based on learned architecture weights. However, the performance estimation ranked by architecture weights in a program derivation graph can be inaccurate due to the co-adaption among architectures via node sharing. Recent work also discovers that relaxed architecture search methods tend to overfit to certain functions that lead to more rapid gradient descent than others [14–17] and thus produce unsatisfying performance.

To overcome this potential disadvantage of differentiable architecture search, our algorithm introduces a search procedure as depicted in Fig. 4. The core idea is that while one super program derivation graph may not be able to model the entire search space accurately, multiple sub program derivation graphs can be used to effectively address the limitation by having each sub graph modeling one part of the search space.

In the search, Algorithm 1 maintains a queue $Q$ of program derivation graphs sorted by their quality that is initialized to $[G]$. Our algorithm measures the

---

**Algorithm 1:** Program Archit. Synthesis

**Input** : Grammar $\mathcal{G}$, Graph expansion depth $d_s$, Top-$N$ parameter
**Output** : Synthesized Program $P$
$G$ contains only the initial nonterminal;
**while** $G$ *contains nonterminals* **do**
    Extend $G$ depth $d_s$ deeper w.r.t. $\mathcal{G}$;
    Optimize $w$ and $\theta$ in $G$ w.rt. Eq. 2;
    Top-$N$ preservation in $G$'s nodes;

$Q := [G]$;
**while** $Q \neq \emptyset$ **do**
    $q := \arg\min_{q \in Q} f(q)$;
    $Q := Q \setminus \{q\}$;
    **if** $q$ *is a well-typed program* **then**
        **return** $q$;
    $u$ is the top-left most node in $q$ with more than one architecture choice;
    **for** *each partial archit.* $f_k^u$ *on* $u$ **do**
        $q' := q[u/f_k^u]$;
        Compute $g(q')$, $h(q')$, $s(q')$;
        $Q := Q \cup \{q'\}$;

---

quality of a program by both its task performance and structure cost. The algorithm dequeues one graph $q$ from $Q$ and extracts the top-most and left-most node $u$ of $q$ that still contains more than one partial architecture for search. As $u$ co-adapts multiple architectures, we separate the entire search space into disjoint partitions by picking each available architecture $f_k^u\left(\alpha_1^{u,k}, \ldots, \alpha_{\eta(f_k^u)}^{u,k}\right)$ from the compound node $u$ and assign a sub program derivation graph to model each partition. The algorithm computes a quality score $s$ for each option of retaining only $f_k^u$ on $u$, denoted as $q[u/f_k^u]$:

$$s(q[u/f_k^u]) = g(q[u/f_k^u]) + h(q[u/f_k^u])$$

The $g(q[u/f_k^u])$ function measures the structure cost of expanding the initial nonterminal up to $u$ (Sec. 2) and $h(q[u/f_k^u])$ is an $\epsilon$-Admissible heuristic estimate of the cost-to-go from node $u$ [18]:

$$h(q[u/f_k^u]) = 1 - F_1(\mathcal{T}_{w^*,\theta^*}[u/f_k^u], D_{val}) \text{ where } w^*, \theta^* = \arg\min_{w,\theta} \mathbb{E}_{i_k,o_k \sim D}[\ell(\mathcal{T}_{w,\theta}[u/f_k^u]), o_k)]$$

where $\mathcal{T}$ encodes the program derivation graph $q$ itself via Equation (3) as a differentiable program whose output is weighted by the output of all complete programs included in $q$, $w$ and $\theta$ are the sets of architecture weights and unknown program parameters in the subgraph rooted at $u$ in $q[u/f_k^u]$. The $h$

function fine-tunes these trainable variables using the training dataset $D$ to provide informed feedback on the contribution to program quality of the choice of only retaining $f_k^u$ on node $u$, measured by the program's $F_1$ score. In practice, to avoid overfitting, we use a separate validation dataset $D_{val}$ to obtain the $F_1$ score. After computing the quality score $s$, we add $q[u/f_k^u]$ back to the queue $Q$ sorted based on $s$-scores. The search algorithm completes when the derivation graph with the least $s$-score from $Q$ is a well-typed program, i.e. each graph node contains only one valid architecture choice. Our architecture selection algorithm is optimal given the admissible heuristic function $h$ — the returned program optimally balances program accuracy and structure complexity among all the programs contained in $G$. The proof is given in Appendix A.

## 4 Experiments

We have implemented Algorithm 1 in a tool named dPads (**d**omain-specific **P**rogram **a**rchitecture **d**ifferentiable **s**ynthesis) [19], and evaluated it on four sequence classification datasets.

### 4.1 Datasets for Evaluation

We partition a dataset to training, validation, and test datasets. dPads uses the training dataset to optimize the architecture weights and program parameters in a program derivation graph. When searching a final program from a converged program derivation graph, we use the validation dataset to obtain the program's $F_1$ score to guide the search. We use the test dataset to obtain the final accuracy and $F_1$ score of a program. Additionally, in training we construct two separate datasets by randomly selecting 60% of a training dataset as $D_\theta$ to optimize program parameters $\theta$ and using the remaining 40% as $D_w$ to train architecture weights $w$ via Equation 2.

**Crim13 Dataset**. The dataset collects social behaviors of a pair of mice. We cut every 100 frames as a trajectory. Each trajectory frame is annotated with an action by behavior experts [20]. For each frame, a 19-dimensional feature vector is extracted including the positions and velocities of the two mice. The goal is to synthesize a program to classify each trajectory to action *sniff* or *no sniff*. In total we have 12404, 3077, and 2953 trajectories in the training set, validation set, and test set respectively.

**Fly-vs-fly Dataset**. We use the *Boy-meets-boy*, *Aggression* and *Courtship* datasets collected in the *fly-vs-fly* environment for monitoring two fruit flies interacting with each other [21]. Each trajectory frame is a 53-dimensional feature vector including fly position, velocity, wing movement, etc. We subsample the dataset similar to [7], which results in 5341 train trajectories, 629 validation trajectories, and 1050 test trajectories. We aim to synthesize a program to classify each trajectory as one of 7 actions displaying aggressive, threatening, and nonthreatening behaviors.

**Basketball Dataset**. The dataset tracks the movement of a basketball, 5 defensive players and 5 offensive players [22]. Each trajectory has 25 frames with each frame as a 22-dimensional feature vector of ball and player position information. We aim to learn a program that can predict which offensive player handles the ball or whether the ball is being passed. In total we have 18000, 2801, and 2693 trajectories in the training set, validation set, and test set respectively.

**Skeletics 152 Dataset**. The dataset [23] contains 152 human pose actions as well as related YouTube Videos subsampled from Kinetics-700 [24]. For each video frame, 25 3-D skeleton points are collected, resulting in a 75-dimensional feature vector per frame. We extract 100 frames from each trajectory to reduce noise. Finally, the training set contains 8721 trajectories, the validation set contains 2184 trajectories, and the test set contains 892 trajectories. We aim to learn a program to classify a pose trajectory as one of 10 actions.

As discussed in Sec. 2, the DSL for each dataset is equipped with a customized library of differentiable and parameterized functions $F_{S,\theta}(x)$. We define these functions in Appendix B. In this paper, we focus on sequence classification benchmarks. However, dPads is a general program synthesis algorithm and is not limited to sequence classification. In Appendix C.6, we evaluate dPads on cryptographic circuit synthesis to demonstrate the generalizability of dPads.

### 4.2 Experiment Setup

To train architecture weights $w$ and unknown program parameters $\theta$ in a differentiable program architecture derivation graph, we use the Adam optimizer [25]. In Algorithm 1, we set $N = 2$

Table 1: Experiment results on the performance of dPads compared with NEAR [7]. All results are reported as the average of runs on five random seeds. Costs of time are set in minutes.

| | Crim13-sniff | | | Fly-vs-fly | | | Bball-ballhandler | | | SK152-10 actions | | |
|---|---|---|---|---|---|---|---|---|---|---|---|---|
| | $F_1$ | Acc. | Time | $F_1$ | Acc. | Time | $F_1$ | Acc. | Time | $F_1$ | Acc. | Time |
| RNN | .481 | .851 | - | .964 | .964 | - | .980 | .980 | - | .414 | .428 | - |
| A*-NEAR | .286 | .820 | 164.92 | .828 | .764 | **243.82** | .940 | .934 | 553.01 | .312 | .315 | 210.23 |
| IDS-BB-NEAR | .323 | .834 | 463.36 | .822 | .750 | 465.57 | .793 | .768 | 513.33 | .314 | .317 | 848.44 |
| dPads | **.458** | .812 | **147.87** | **.887** | .853 | 348.25 | **.945** | .939 | **174.68** | **.337** | .337 | **162.70** |

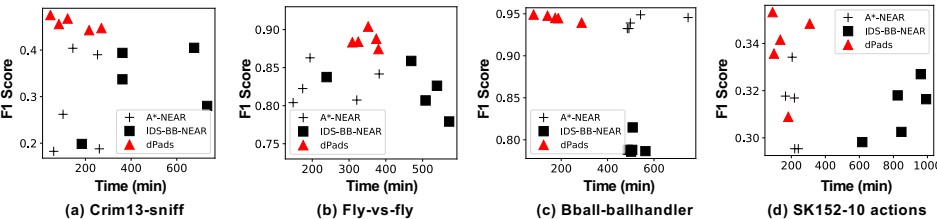

(a) Crim13-sniff  (b) Fly-vs-fly  (c) Bball-ballhandler  (d) SK152-10 actions

Figure 5: Experiment results on the Crim13, Fly-vs-fly, Basketball and SK152 datasets over five random seeds. $x$ axis refers to costs of time recorded in minutes and $y$ axis refers to $F_1$ scores.

for top-$N$ preservation and set graph expansion depth $d_s$ to 2. For evaluation, we compare dPads with the state-of-the-art program learning algorithms A*-NEAR and IDS-BB-NEAR [7]. We only report a comparison with NEAR because NEAR significantly outperforms other program learning methods based on top-down enumeration, Monte-Carlo sampling, Monte-Carlo tree search, and genetic algorithms [7]. All experiments were performed on Intel 2.3-GHz Xeon CPU with 16 cores, equipped with an NVIDIA Quadro RTX 6000 GPU. More experiment settings including learning rates and training epochs are given in Appendix C.1 and C.2.

## 4.3 Experiment Results

For a fair comparison with NEAR [7], for any of the four datasets, all tools search over the same DSL. We use random seeds 0, 1000, 2000, 3000, 4000 and report average $F_1$ scores, accuracy rates and execution times for both methods. We also report the results achieved using a highly expressive RNN baseline which provides a task performance upper bound on $F_1$-scores and accuracy.

Table 1 shows the experiment results. On both the Crim13 and SK152 datasets, dPads outperforms A*-NEAR and IDS-BB-NEAR achieving higher $F_1$ scores and using much less time consumption. dPads also achieves competitive accuracy with NEAR on Crim13. On the Basketball dataset, although dPads achieves a bit higher $F_1$ score, the architectures synthesized by dPads and A*-NEAR are exactly the same. However, dPads takes 70% less time to get the result. While A*-NEAR completes the search faster than dPads on Fly-vs-fly, the program architecture synthesized by dPads leads to a program with a much better $F_1$ score and higher accuracy. More quantitative analyses of the experiment results are given in Appendix C.3.

We visualize the results of dPads and NEAR in terms of $F_1$ scores ($y$ axis) and running times ($x$ axis) on the 5 random seeds in Fig. 5 where red triangles refer to the results of dPads, and black plus marks and rectangles refer to the results of A*-NEAR and IDS-BB-NEAR. dPads consistently outperforms NEAR in achieving higher $F_1$ scores with less computation and is closer to the RNN baseline.

Although the RNN baseline provides better performance, dPads learns programs that are more interpretable. Fig. 6 depicts the best programs synthesized by dPads on Crim13 and SK152 (among all the 5 random seeds). The program for Crim13 has a simple architecture and achieves a high $F_1$ score 0.475 (only 0.006 less than the RNN result). It invokes two $F_{S,\theta}$ library functions: PositionAffine and DistanceAffine. This program is highly human-readable: it evaluates the likelihood of "sniff" by applying a position bias and if the distance between two mice is small they are doing a "sniff". The programmatic classifier for SK152 achieves an $F_1$ score 0.35 which is close to the RNN baseline. It uses the arm and leg positions of a 3-D skeleton to complete a human-action classification. We show more examples about the interpretability of programs learned by dPads in Appendix C.5.

```
Map(                           SlideWindowAvg(Add(
  Multiply(                        Multiply(LegsAffine_{θ_1}(x_t),
    PositionAffine_{θ_1}(x_t)),                 LegsAffine_{θ_2}(x_t)),
    DistanceAffine_{θ_2}(x_t)))  x     Multiply(ArmsAffine_{θ_3}(x_t),
                                                ArmsAffine_{θ_4}(x_t))))  x
```

Figure 6: Synthesized Programs for Crim13-sniff (left) and SK152-10 actions (right).

Table 2: Ablation study on the importance of node sharing and iterative graph unfolding as two optimization strategies in dPads. All results are reported as the average of runs on five random seeds. Costs of time are set in minutes. OOM represents an out-of-memory error.

| Variants of dPads | Crim13-sniff | | | Fly-vs-fly | | | Bball-ballhandler | | | SK152-10 actions | | |
| | $F_1$ | Acc. | Time | $F_1$ | Acc. | Time | $F_1$ | Acc. | Time | $F_1$ | Acc. | Time |
|---|---|---|---|---|---|---|---|---|---|---|---|---|
| dPads w/o Node Sharing | .453 | .800 | 334.93 | - | - | >1440 | - | - | >1440 | .321 | .322 | 252.81 |
| dPads w/o Graph Unfolding | .449 | **.818** | 280.67 | - | - | OOM | .848 | .832 | 348.09 | **.348** | **.346** | 273.95 |
| dPads in full | **.458** | .812 | **147.87** | **.887** | **.853** | 348.25 | **.945** | **.939** | 174.68 | .337 | .337 | **162.70** |

## 4.4 Ablation Studies

We introduce two more baselines to study the importance of node sharing and iterative graph unfolding. The first baseline does not use node sharing to reduce the size of a program derivation graph but still performs iterative graph unfolding. The second baseline directly expands a program derivation graph to the maximum depth but still applies node sharing. We report the comparison results over 5 random runs in Table 2. Without the two optimizations, limited by the size of GPU memory, dPads may either time-out or encounter out-of-memory error when searching programs that need deep structures to ensure high accuracy. This is because the size of a program derivation graph grows exponentially large with the height of program abstract syntax trees and the number of DSL production rules. Moreover, while being more complete, training without these two optimizations does not necessarily produce better results even when there is not OOM or timeout. For example, on Basketball, dPads achieves .945 $F_1$ score. dPads without iterative graph unfolding only obtains .848 $F_1$ score. We suspect this is because the program derivation graph without top-$N$ preservation and iterative unfolding is more difficult to train as it contains significantly more parameters.

We further investigate the effect of the top-$N$ preservation strategy in program architecture synthesis (Sec. 3.2) and its impact on searching optimal programs (Sec. 3.3). We set $N$ to 1, 2, 3 respectively and study how dPads responds to these changes. Table 3 summarizes the average results of $F_1$ scores, accuracy rates, time costs, and the standard deviations of these results. When $N = 1$, dPads extracts final programs greedily from optimized program derivation graphs without conducting further search. There is a significant decrease in time consumption compared with $N = 2$. However, dPads in this condition achieves less $F_1$ scores and the results have higher variances, which suggests that architecture weights learned using only differentiable synthesis overfit to sub-optimal programs. dPads gets *similar* $F_1$ scores when setting $N = 3$ compared to $N = 2$ but consumes more time. It even times-out on the Basketball dataset while searching an optimal program from the converged program derivation graph, since $N = 3$ incurs a much larger search space. This result confirms that scaling discrete program search to large architecture search spaces is challenging. dPads addresses this fundamental limitation by leveraging differentiable search of program architectures to significantly prune away unlikely search directions. Therefore, it suffices to set $N = 2$ in our experiments to balance search optimality and efficiency. Additional ablation study results are given in Appendix C.4. The limitations of dPads are discussed in Appendix D.1.

## 5 Related Work

**Program Synthesis.** Tasks in program synthesis aim to search for programs in a DSL to satisfy a specification over program inputs and outputs. There is also a growing literature on applying deep learning methods to guide the search over program architectures [6, 26–34]. There exist efforts that extend this line of research to program synthesis from noisy data [1, 2, 35–37, 3, 4]. These approaches either require a detailed hand-written program template or simply enumerate the discrete

Table 3: Ablation study on the value of $N$ for the top-$N$ preservation strategy used in dPads. All results are reported as the average of runs on five random seeds. Costs of time are set to minutes.

| | $N$ | dPads | | | | |
| --- | --- | --- | --- | --- | --- | --- |
| | | $F_1$ | Acc. | Time | Std. $F_1$ | Std. Acc. |
| **Crim13-sniff** | 1 | .272 | .627 | 50.85 | .111 | .218 |
| | 2 | .458 | .812 | 147.87 | .014 | .008 |
| | 3 | .450 | .811 | 441.12 | .025 | .008 |
| **Fly-vs-fly** | 1 | .769 | .716 | 95.36 | .052 | .062 |
| | 2 | .887 | .853 | 348.25 | .010 | .006 |
| | 3 | .866 | .818 | 620.48 | .017 | .039 |
| **Bball-ballhandler** | 1 | .808 | .785 | 41.14 | .042 | .045 |
| | 2 | .945 | .939 | 174.68 | .004 | .004 |
| | 3 | - | - | > 1440 | - | - |
| **SK152-10 actions** | 1 | .310 | .310 | 40.34 | .020 | .024 |
| | 2 | .337 | .337 | 162.70 | .017 | .017 |
| | 3 | .336 | .338 | 609.14 | .011 | .010 |

space of program architectures permitted by a DSL. Additionally, most of these literature methods build models that are trained using corpora of synthesis problems and solutions, which are not available in our setting. The most closest work to our technique includes [38, 7] that enumerate the space of program architectures prioritizing search directions with feedback from machine learning models. Specifically, Lee et al. [38] uses a probabilistic model (trained from a synthesis problem corpus) to guide an A* search over discrete program syntax trees and NEAR [7] uses neural networks to approximate missing expressions in a partial program whose $F_1$ score serves as an admissible heuristic to guide an A* search again over discrete program syntax trees. As opposed to these efforts, our method more efficiently conducts program synthesis in a continuous relaxation of the discrete space of language grammar rules and only searches the optimal program in a much reduced search space after differentiable architecture synthesis for addressing its gradient bias problem [14].

**Differentiable Architecture Search.** Neural architecture search has attracted much interest as a promising approach to automate deep learning tasks [39–42]. Particularly, our program architecture synthesis algorithm is inspired by DARTS [10]. This method uses a composition of softmaxes over all possible candidate operations between a fixed set of neural network nodes to relax the neural architecture search space. Various methods further improve neural architecture search efficiency and accuracy [14–17, 43, 44]. Applying this line of algorithms to program synthesis is challenging because the space of program architectures is much richer. Different operations take different number and types of inputs/outputs and may only be available at different points of a program. There is also no fixed bound on the number of program expressions. By relaxing the discrete search space of language grammar rules with node sharing and iterative unfolding of program derivation graphs, our method addresses the aforementioned challenges. To the best of our knowledge, this is the first approach that applies differentiable architecture search to program synthesis.

## 6 Conclusions

This paper presents a novel differentiable approach to program synthesis. With gradient descent, our method learns the probability distribution of program architectures induced by the context-free grammar of a DSL in a continuous relaxation of the discrete space of language grammar rules. This allows the synthesis algorithm to efficiently prune away unlikely program derivations to discover optimal program architectures. We have instantiated differentiable program architecture synthesis with effective optimization strategies including node sharing and iterative graph unfolding, scaling it to real-world sequence classification tasks. Experiment results demonstrate that our algorithm substantially outperforms state-of-the-art program learning approaches.

Programmatic models in high-level DSLs are a powerful abstraction for summarizing discovered knowledge from data in a human-interpretable way. Programmatic models incorporate inductive bias through structured symbolic primitives in a DSL and open opportunities for programmers to influence the semantic meaning of learned programs. However, the programming biases in a DSL may also leave opportunities to attack on the security and fairness of a learned model. One direction for future work is to apply formal program reasoning to enhance the trustworthiness of programmatic models.

## Acknowledgments and Disclosure of Funding

We thank the anonymous reviewers for their comments and suggestions. This work was supported by NSF Award #CCF-2124155 and the DARPA Symbiotic Design for Cyber Physical Systems program.

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
