# A    Optimality of the search procedure in Algorithm 1

In Algorithm 1, once we have an optimized program derivation graph $G$, due to the top-$N$ preservation strategy, each node retains a small number of partial architectures. Algorithm 1 maintains a queue $Q$ of program derivation graphs that is initialized to $[G]$. The algorithm dequeues one graph $q$ from $Q$ and extracts the top-most and left-most node $u$ of $q$ that contains more than one partial architecture for search. It enumerates each available partial architecture $f_k^u\left(\alpha_1^{u,k}, \ldots, \alpha_{\eta(f_k^u)}^{u,k}\right)$ on $u$ and computes an $s$-score for each option of retaining only $f_k^u$ on $u$, denoted as $q[u/f_k^u]$. We define

$$s(q[u/f_k^u]) = g(q[u/f_k^u]) + h(q[u/f_k^u])$$

The $g(q[u/f_k^u])$ function measures the structure cost of expanding the initial nonterminal up to $u$ and $h(q[u/f_k^u])$ is an $\epsilon$-Admissible heuristic estimate of the cost-to-go from node $u$ [7, 18] for A\* search:

$$\begin{aligned}
h(q[u/f_k^u]) = & 1 - F_1(\mathcal{T}_{w^*,\theta^*}[u/f_k^u], D_{val}) \\
& \text{where } w^*, \theta^* = \arg\min_{w,\theta} \mathbb{E}_{i_k,o_k \sim D}[\ell(\mathcal{T}_{w,\theta}[u/f_k^u]), o_k)]
\end{aligned} \tag{4}$$

where $\mathcal{T}$ encodes the program derivation graph $q$ itself via Equation (3) as a differentiable program whose output is weighted by the output of all complete programs included in $q$, $w$ and $\theta$ are the sets of architecture weights and unknown program parameters in the subgraph rooted at $u$ in $q[u/f_k^u]$. The $h$ function fine-tunes these trainable variables using the training dataset $D$ to provide informed feedback on the contribution to program quality of the choice of retaining $f_k^u$ on node $u$, measured by $F_1$ score. In practice, to avoid overfitting, we use a separate validation dataset $D_{val}$ to obtain the $F_1$ score. After computing the quality score $s$, we add $q[u/f_k^u]$ back to the queue $Q$ sorted based on $s$-scores. The search algorithm completes when the derivation graph with the least $s$-score from $Q$ is a well-typed program, i.e. each graph node contains only one valid architecture choice.

We aim to prove that our search algorithm is optimal given the admissible heuristic function $h$. When multiple solutions exist in $G$ (when the top-$N$ parameter is greater than 1), the algorithm finds an optimal solution. Among all the programs contained in $G$, the synthesized program optimally balances program accuracy and structure complexity.

We note that our search algorithm is a variant of A\* search by by interpreting $g(q[u/f_k^u])$ as cost-so-far and $h(q[u/f_k^u])$ as heuristic cost-to-go. A\* search is optimal given admissible heuristics. We shows that under heuristics that are $\epsilon$-admissible, our search algorithm returns solutions that at most an additive constant $\epsilon$ away from the optimal solution.

Firstly, we prove that that our heuristic function $h$ is $\epsilon$-admissible. Let a completion of a partial architecture $q[u/f_k^u]$ be a (complete) architecture $\tilde{q}[u/f_k^u]$ obtained by retaining only one partial architecture on any node of $q$. The cost-to-go at $q[u/f_k^u]$ is given by:

$$J(q[u/f_k^u]) = \min_{\tilde{q}[u/f_k^u]} c(\tilde{q}[u/f_k^u]) - c(q[u/f_k^u]) + 1 - F_1(\tilde{q}[u/f_k^u], D_{val})$$

where the structural cost $c(q)$ is the sum of the costs of the grammatical rules used to construct $q$ excluding any nodes with more than 1 partial architectures (i.e. unexplored nodes in search).

The optimization in Equation (4) may only converge to a local minimum. However, since our relaxation of the search space for Equation (4) includes any possible program permitted by $q$, there must exist architecture weights $w^*$ and program parameters $\theta^*$ such that

$$\forall \tilde{q}[u/f_k^u].\ 1 - F_1(\mathcal{T}[u/f_k^u], w^*, \theta^*, D_{val}) \leq 1 - F_1(\tilde{q}[u/f_k^u], D_{val}) + \epsilon$$

Thus we have:

$$\begin{aligned}
h(q[u/f_k^u]) \leq\ & 1 - F_1(\mathcal{T}[u/f_k^u], w^*, \theta^*, D_{val}) \\
\leq\ & \min_{\tilde{q}[u/f_k^u]} 1 - F_1(\tilde{q}[u/f_k^u], D_{val}) + \epsilon \\
\leq\ & \min_{\tilde{q}[u/f_k^u]} c(\tilde{q}[u/f_k^u]) - c(q[u/f_k^u]) + 1 - F_1(\tilde{q}[u/f_k^u], D_{val}) + \epsilon \\
\leq\ & J(q[u/f_k^u]) + \epsilon
\end{aligned} \tag{5}$$

In other words, $h(q[u/f_k^u])$ is $\epsilon$-admissible as for a fixed constant $\epsilon > 0$, $h$ is an $\epsilon$-admissible heuristic function over architectures such that $h(q[u/f_k^u]) \leq J(q[u/f_k^u]) + \epsilon$ for any partial architecture $f_k^u$ on $u$.

Table 4: $F_{S,\theta}(x)$ for the Crim13 dataset.

| Extract Feature | Dimension |
|---|---|
| Position | 0, 1, 2, 3 |
| Distance | 4 |
| Distance Change | 5 |
| Velocity | 11, 12, 13, 14 |
| Acceleration | 15, 16, 17, 18 |
| Angle | 6, 7, 10 |
| Angle Change | 8, 9 |

Table 5: $F_{S,\theta}(x)$ for the Fly-vs-fly dataset.

| Extract Feature | Dimension |
|---|---|
| Linear | 17, 25 |
| Angular | 18, 26, 27 |
| Positional | 24, 28 |
| Ratio | 22, 23 |
| Wing | 19, 20, 21 |

Table 6: $F_{S,\theta}(x)$ for the Basketball dataset.

| Extract Feature | Dimension |
|---|---|
| Ball | 0, 1 |
| Offense | 2, 3, 4, 5, 6, 7, 8, 9, 10, 11 |
| Defence | 12, 13, 14, 15, 16, 17, 18, 19, 20, 21 |

$\epsilon$-**Optimality.** Based on the $\epsilon$-admissible of heuristic function $h(q[u/f_k^u])$, we prove that the variant of $A^*$ search in Algorithm 1 results in a synthesized program that is at most $\epsilon$ away from the optimal solution contained in the search graph. Suppose that Algorithm 1 returns a program $P^r$ that does not have the optimal cost $C^*$. Then there must exist a program derivation graph $q^*$ in the queue $Q$ of Algorithm 1 that contains the architecture of the optimal program $P^*$. Due to Equation 5 and the fact that $Q$ is sorted, the $s$-score of $P^r$ satisfies:

$$\begin{aligned} s(P^r) &\leq s(q^*[u^*/f^*]) \\ &= g([q^*[u^*/f^*]) + h(q^*[u^*/f^*]) \\ &\leq g(q^*[u^*/f^*]) + J(q^*[u^*/f^*]) + \epsilon \\ &\leq C^* + \epsilon \end{aligned}$$

where $u^*$ is the top-most and left-most node of $q^*$ and $f^*$ is the optimal partial architecture to retain on node $u^*$ to get the optimal program $P^*$. In other words, we have established an upper bound on the path cost of the returned synthesized program $P^r$.

## B  Context-free Grammar Details

We use the context-free grammar in Fig. 1 for synthesizing programmatic classifiers for all the datasets in our experiment. For each dataset, similar to NEAR [7], we customize parameterized functions $F_{S,\theta}(x)$ that extract a vector consisting of a predefined subset $S$ of the dimensions of an input data item $x$ and pass the extracted vector through a linear function with trainable parameters $\theta$. We disclose the details of $F_{S,\theta}(x)$ for each dataset below.

**Crim13 Dataset.** As shown in Table 4, we define 7 feature extraction functions $F_{S,\theta}(x)$ for the Crim13 Dataset. For location information, XY positions of a pair of mice and the distance between them are recorded. Additionally, *distance change* measures the distance difference for each two consecutive frames. To track movement information, velocity and acceleration of a pair of mice are extracted in X and Y dimensions respectively. Besides, we also include the information on *angle* and *angle change*. The former contains the two relative directions between a pair of mice (one for mouse 1 relative to mouse 2, another for mouse 2 relative to mouse 1) and the difference between the two relative directions. *Angle change* represents the change of the two relative directions over time. More information on the dimensions of this dataset can be found in [20].

**Fly-vs-fly Dataset.** Table 5 shows the feature functions $F_{S,\theta}(x)$ we extract for the Fly-vs-fly dataset. Although the feature vectors of the dataset have 53 dimensions, we find the 5 feature functions in the table are sufficient to obtain high accuracy and $F_1$-scores. The *linear* feature function captures the values of velocity and distance between two flies. The *Angular* feature function extracts the value of angle velocity, relative angle between flies and facing angle of each fly. The feature function *Positional* captures the distance over a relative object and fly legs. The feature function *ratio* extracts the body ratio of two flies. The feature function *wing* extracts the angles and lengths of fly wings.

Table 7: $F_{S,\theta}(x)$ for the SK152 dataset.

| Extract Feature | Point |
|---|---|
| Arms | 2, 3, 4, 5, 6, 7 |
| Legs | 8, 9, 10, 11, 12, 13, 14, 19, 20, 21, 22, 23, 24 |
| Faces | 0, 1, 15, 16, 17, 18 |

Table 8: Dataset details

| Dataset | feature dim. | category num. | max seq. len. | # train | # valid | # test |
|---|---|---|---|---|---|---|
| **Crim13** | 19 | 2 | 100 | 12404 | 3077 | 2953 |
| **Fly-vs-fly** | 53 | 7 | 300 | 5341 | 629 | 1050 |
| **Basketball** | 22 | 6 | 25 | 18000 | 2801 | 2693 |
| **SK152** | 75 | 10 | 100 | 8721 | 2184 | 892 |

**Basketball Dataset.** As shown in table 6, we define three feature extraction functions for the Basketball dataset, extracting the positions of the basketball, 5 offensive players, and 5 defensive players. All positions are expressed in X and Y coordinates.

**SK152 Dataset.** This dataset uses a total of 25 points to capture human skeletons. Each point is recorded using XYZ coordinates. We define three customized feature functions *arms*, *legs* and *face*, each of which extracts a subset of the 25 features.

## C  Experiment Details

We provide more details about our experiment in this section.

### C.1  Training Details

**Datasets.** Table 8 gives the full details of the four datasets used for evaluation. NEAR [7] does not release the Fly-vs-fly and Basketball datasets used to obtain the results in its paper. We sample these datasets following the guidance provided in [7]. Therefore, the datasets used in the evaluation of this paper are not completely equivalent to that of [7].

**Structure Cost.** To penalize complex program architectures, we implement the structure cost function $g$ similar to NEAR [7]. Let each grammar rule $r$ have a non-negative real-valued cost c(r). The structural cost of a (partial) architecture is the sum of the costs of the multi-set of rules used to create the architecture.

$$g(q[u/f_k^u]) = \beta \cdot \sum_{r \in q[u/f_k^u]} c(r)$$

Importantly, the above formula only counts the grammar rules used to expand the initial nonterminal up to node $u$ (recall that $u$ is the top-most and left-most node of $q$ that contains more than one partial architecture i.e. unexplored nodes are excluded). To balance the structure cost and performance of a programmatic classifier, we set the cost penalty parameter $\beta$ to be 0.01 for both dPads and NEAR. In practice, we set $c(r) = 1$ for any grammar rule $r$. For a complete program, the $g$ function essentially counts the number of grammar rules used to derived the program (timed with $\beta$).

### C.2  Hyperparameters

**RNN Baseline.** The RNN baseline policies are 1-layer LSTMs. Table 9 introduces the hyperparameters used for training the RNN baselines. In general, the RNN baselines perform better than the synthesized programmatic classifiers because their richer structures allow for better data fitting at the cost of less interpretability. In the experiments, we use the cross-entropy loss to optimize classifier accuracy for both dPads and baselines.

Several other baselines that we considered include (1) Top-down enumeration that synthesizes and evaluates complete programs in order of increasing complexity measured using the structural cost, (2) Monte-Carlo sampling that constructs complete programs by sampling rules (edges) with

Table 9: Hyperparameters set for the RNN baseline.

| Dataset | # LSTM units | # epochs | learning rate | batch size |
|---|---|---|---|---|
| **Crim13** | 100 | 50 | 0.001 | 50 |
| **Fly-vs-fly** | 80 | 40 | 0.00025 | 30 |
| **Basketball** | 64 | 15 | 0.01 | 50 |
| **SK152** | 75 | 30 | 0.01 | 50 |

Table 10: Hyperparameters set for Differentiable Architecture Search and Selection in dPads.

| Dataset | Architecture Search | Architecture Selection | | | Batch Size |
|---|---|---|---|---|---|
| | learning rate | graph_epoch | prog_epoch | learning rate | |
| **Crim13** | 0.001 | 6 | 6 | 0.001 | 200 |
| **Fly-vs-fly** | 0.0005 | 6 | 6 | 0.0005 | 200 |
| **Basketball** | 0.001 | 4 | 6 | 0.02 | 50 |
| **SK152** | 0.01 | 4 | 6 | 0.01 | 200 |

probabilities proportional to their structural costs where the next node to expand along a path has the best average performance of samples that descended from that node, (3) Monte-Carlo tree search (MCTS) that traverses the search graph of programs using the UCT selection criteria, where the value of a node is inversely proportional to the cost of its children, and (4) Genetic algorithm that uses crossover, selection, and mutation operations to evolve a population of programs over a number of generations. Unlike dPads, these baselines perform program architecture search in the discrete space of DSL grammar rules. We do not include these baselines in the experiment section because NEAR significantly outperforms them [7]. Therefore, it suffices to compare dPads solely with NEAR.

**dPads**. In the evaluation of dPads, we set graph expansion depth $d_s$ (a parameter of Algorithm 1) as 2 for all the four datasets. NEAR uses the number of grammar production rules applied to construct a program to upper-bound the search space for program learning. The largest number of production rules allowed for a synthesized program is 8 in NEAR. Instead, we set the max depth of the abstract syntax tree of a dPads's synthesized program as 4. Compare to the search space of NEAR, some programs included in the search space of dPads even need more than 10 production rules to expand from the initial nonterminal.

For top-$N$ preservation in a program derivation graph, the goal is to retain top-$N$ program architectures as children for each partial architecture on the node's parent, which are defined to be those assigned with higher weights on the node's incoming edge in the previous graph unfolding iteration. We set $N = 2$ in our experiments. In practice, we find that a heuristic strategy that *iteratively* prunes candidate partial architectures on a program derivation graph node and fine-tunes the derivation graph at then end of each *iteration* is more efficient than directly retaining top-$N$ architectures. In a graph unfolding iteration, after optimizing the architecture weights and unknown program parameter over an entire program derivation graph for several epochs until the increase of the $F_1$ score of the whole graph is less than 1%, on each node we retain 4 program architectures as children for each partial architecture on the node's parent, then we retrain the program derivation graph for several epochs again until the increase of $F_1$ score is less than 1% and on each node we retain 3 program architectures for each partial program architectures on the node's parent, and finally we apply such a process again to retain only 2 program architectures to get the desired top-2 preservation in the program derivation graph.

Table 10 presents the hyperparameters used to train dPads programmatic models. The learning rates for differentiable architecture search on a program derivation graph and architecture selection from the converged program derivation graph may be different. For architecture selection, the number of graph_epoch refers to the number of epochs that is used to optimize the heuristic function $h$ (Equation 4) when a program derivation graph has nodes containing more than one candidate architectures. Here we need to optimize both architecture weights and program parameters. The number of prog_epoch refers to the number epochs that is used to fine-tune the accuracy of a program derivation graph when every node of the graph contains exactly one architecture, i.e., the graph is the abstract syntax tree of a valid program. For Crim13 and Fly-vs-fly, both graph_epoch and prog_epoch are set to 6. For Basketball and SK152, graph_epoch is set to 4 and prog_ epoch is set to 6.

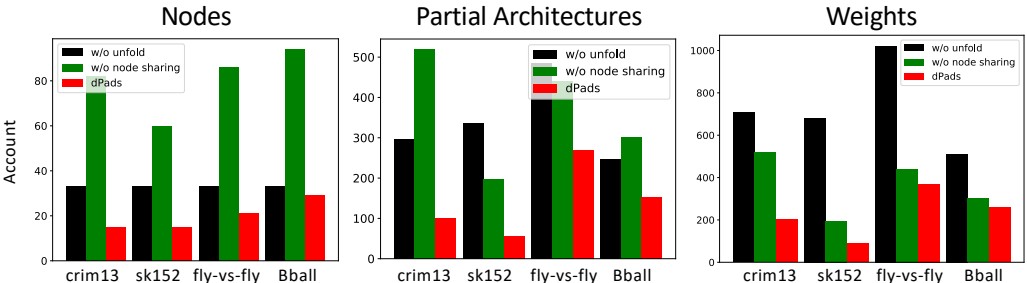

Figure 7: Results of quantifying the number of graph nodes, the total number of partial architectures (DSL functions) hosted by the nodes, and the total number of architecture weights on graph edges on program derivation graphs generated by dPads and its variants for each of the datasets.

## C.3  Additional Experiment Results

**Search Space Reduction.** Fig. 7 quantifies the program derivation graph reduction by node sharing and iterative unfolding. We show for each benchmark the number of nodes, the total number of partial architectures (DSL functions) hosted by the nodes, and the total number of architecture weights to train on its program derivation graph. We only show the results of the deepest program derivation graph generated by dPads on each benchmark. Notice that dPads without iterative graph unfolding and dPads without node sharing generate program derivation graphs that are significantly larger.

On Fly-vs-fly, without iterative graph unfolding, the program derivation graph has 33 nodes shared by 485 partial architectures, and a total of 1021 architecture weights to train. Directly applying dPads to train under this setting encounters an out-of-memory (OOM) exception (Table 2). In contrast, with iterative graph unfolding, the deepest graph generated by dPads has only 21 nodes shared by 220 partial architectures, and a total of 370 architecture weights to train. dPads converges in less than 360 mins (Table 2).

On Basketball, without node sharing, the deepest program derivation graph generated by dPads has 94 nodes hosting 302 partial architectures, and a total of 301 architecture weights to train. If directly running dPads under this setting, the search procedure (Sec. 3.3) times-out (Table 2). In contrast, with node sharing, the program derivation graph reduces to only 29 nodes shared by 152 partial architectures, and a total of 259 architecture weights. dPads converges in less than 180 mins (Table 2).

**Program Sizes.** Table 11 reports the number of grammar rules used to construct synthesized programs for each benchmark. Compared with NEAR, dPads tends to synthesize more complex programs that are necessary to ensure higher $F_1$ scores.

On Fly-vs-fly, although NEAR runs faster, it only finds programs derived by 2.8 production rules but dPads finds much deeper programs derived by 6.8 production rules. Consequently, for this benchmark, dPads achieves much higher accuracy and $F_1$ scores. A similar trend can be observed for the results on Crim13 and SK152. dPads learns program classifiers for Crim13 constructed using 10.2 grammar rules averagely. These classifiers achieve an average of 0.46 $F_1$ scores (on the test dataset). The shallower classifiers learned by NEAR achieve an average of 0.33 $F_1$ scores constructed by 5.8 grammar rules averagely. While the dPads classifiers are more complex, the total $s$-scores of dPads classifiers $(1 - F_1(p, D_{val}) + \beta \cdot \sum_{r \in AST(p)} c(r)$ where $p$ is a classifier), i.e. synthesized programs' classification error plus architecture cost (the search objective), are still lower than that of the programs learned by NEAR. In other words, dPads better balances structure costs and performance. On Basketball, both tools find programs with 8 production rules. In this case, the architecture spaces searched by the two tools are roughly equivalent, but dPads is 3 times faster.

The experiment results consistently demonstrate that dPads's gradient-based architecture search is much more efficient than NEAR's enumeration-based strategy. Moreover, NEAR uses neural models to estimate the performance of a partially expanded program. Our experiments find that due to overfitting or underfitting, such a neural model may be biased on a particular program. For example, on Fly-vs-fly, due to the biased estimation, NEAR stops searching the architecture space deeper than that contains programs derived by only 2.8 production rules. In contrast, (1) dPads uses a sub program derivation graph to estimate the performance of a partially expanded program that can

Table 11: The average numbers of grammar rules (#R) used to construct synthesized programs together with the programs' $F_1$ scores. All results are reported as the average of runs on five random seeds. Costs of time are set to minutes.

| | Crim13-sniff | | | Fly-vs-fly | | | Bball-ballhandler | | | SK152-10 actions | | |
|---|---|---|---|---|---|---|---|---|---|---|---|---|
| | $F_1$ | #R | Time | $F_1$ | #R | Time | $F_1$ | #R | Time | $F_1$ | #R | Time |
| A*-NEAR | .286 | 6.8 | 164.92 | .828 | 2.8 | **243.82** | .940 | 8.0 | 553.01 | .312 | 4.2 | 210.23 |
| IDS-BB-NEAR | .323 | 5.8 | 463.36 | .822 | 2.4 | 465.57 | .793 | 7.0 | 513.33 | .314 | 4.4 | 848.44 |
| dPads | **.458** | 10.2 | **147.87** | **.887** | 6.8 | 348.25 | **.945** | 8.0 | **174.68** | **.337** | 7.6 | **162.70** |

Table 12: Standard deviations of $F_1$ scores, accuracy rates, and the number of grammar rules used to construct synthesized programs. All results are reported as the average of runs on five random seeds.

| | Crim13 | | | Fly-vs-fly | | | Basketball | | | SK152 | | |
|---|---|---|---|---|---|---|---|---|---|---|---|---|
| | $F_1$ | Acc. | Rules | $F_1$ | Acc. | Rules | $F_1$ | Acc. | Rules | $F_1$ | Acc. | Rules |
| A*-NEAR | .107 | .007 | 3.27 | .025 | .016 | 1.10 | .007 | .009 | 0.0 | .017 | .018 | 0.45 |
| IDS-BB-NEAR | .086 | .008 | 2.39 | .030 | .025 | 0.89 | .012 | .015 | 0.0 | .012 | .010 | 0.54 |
| dPads | .013 | .008 | 5.31 | .011 | .006 | 1.10 | .004 | .004 | 0.0 | .017 | .017 | 0.55 |

provide more accurate assessment due to the graph's syntax resemblance to a valid program; (2) dPads only uses neural models to provide "contrastive" performance estimation for a set of programs sharing nodes in a program derivation graph when iteratively unfolding the graph. As a result, on Fly-vs-fly, dPads searches the architecture space much deeper containing programs derived by 6.8 production rules. Even dPads typically searches much deeper, it often runs faster than NEAR.

**Standard Deviations.** We present the standard deviations of accuracy, $F_1$-scores, and numbers of production rules used to construct learned programs in Table 12. The results confirms the observation in [7] that NEAR has a higher variance in $F_1$ scores for CRIM13. dPads is more stable on this dataset.

**Further Comparison with NEAR.** In the comparison with NEAR, we obtain the NEAR results following the hyperparameters defined in [7]. The only exception is that for Crim13 and Fly-vs-fly, we modify the batch size for NEAR program learning to be 200 while keeping the other hyperparameters unmodified to get the results. This is for ensuring a fair comparison of running times with dPads that sets batch size to 200. In this section, we report the results of NEAR on Crim13 and Fly-vs-fly using exactly the same hyperparameters as reported in [7], setting batch size to 50 and 30 respectively, and show the results in Table 13. In this setting, A*-NEAR and IDS-BB-NEAR get higher $F_1$ scores that are, however, still lower than that of dPads. Moreover, NEAR costs significantly longer time in search. Thus, dPads outperforms NEAR in this setting as well.

**Comparison with Enumerative Program Synthesis.** Besides comparing with NEAR and RNN (as in Table 1), we also compared dPads with an enumeration strategy that synthesizes and evaluates complete programs in order of increasing complexity. This strategy is widely used in program synthesis tasks. We set the running time of the enumeration strategy twice as long as dPads's synthesis time. As shown in Table 14, dPads outperforms enumeration strategy on all benchmarks. Although enumeration gets higher accuracy on Cim13, it underfits this unbalanced dataset as the $F_1$ score is much lower than dPads. We have also tried a Monte-Carlo tree search strategy but found that its performance is even worse than the simple enumeration strategy on our benchmarks.

## C.4 Additional Ablation Study Results

Other than top-$N$ preservation and iteratively unfolding program derivation graphs, we explored other pruning approaches, including reserving the *top-$N$ programs* across the entire search graph. We also considered another pruning algorithm which we refer to as *First Compare First Unfold* (FCFU). Unlike dPads that separates top-$N$ preservation and iterative graph unfolding with the search algorithm in Sec. 3.3, FCFU mixes the two optimizations with the search procedure. It manages a priority queue of program derivation graphs that is initialized to the simplest graph with the initial nonterminal only. After training converges on a program derivation graph from the priority queue, FCFU decomposes the graph into several sub graphs by separating the co-adapted architectures on the top-left most compound node on the graph. On each sub graph, that node contains only one

Table 13: Experiment results of NEAR following the exact hyperparameter setting specified in [7]. All results are reported as the average of runs on five random seeds. Costs of time are set to minutes.

| | Crim13-sniff | | | Fly-vs-fly | | |
|---|---|---|---|---|---|---|
| | $F_1$ | Acc. | Time(mins) | $F_1$ | Acc. | Time(mins) |
| A*-NEAR | .304 | .824 | 519.60 | .873 | .827 | 1208.92 |
| IDS-BB-NEAR | .328 | .840 | 1106.28 | - | - | > 1440 |
| dPads | **.458** | .812 | **147.87** | **.887** | .853 | **348.25** |

Table 14: Experiment results on comparing dPads with an enumeration-based synthesis strategy. The running time of the enumeration strategy is set twice as long as the synthesis time of dPads. All results are reported as the average of runs on five random seeds. Costs of time are set to minutes.

| | Crim13 | | Fly-vs-fly | | Basketball | | SK152 | |
|---|---|---|---|---|---|---|---|---|
| | $F_1$ | Acc. | $F_1$ | Acc. | $F_1$ | Acc. | $F_1$ | Acc. |
| Enumeration | .294 | .856 | .850 | .774 | .795 | .767 | .288 | .284 |
| dPads | .458 | .812 | .887 | .853 | .945 | .939 | .337 | .337 |

Table 15: Ablation study on various pruning methods. All results are reported as the average of runs on five random seeds. Costs of time are set to minutes.

| | Crim13 | | | Fly-vs-fly | | | Basketball | | | SK152 | | |
|---|---|---|---|---|---|---|---|---|---|---|---|---|
| | $F_1$ | Acc. | Time | $F_1$ | Acc. | Time | $F_1$ | Acc. | Time | $F_1$ | Acc. | Time |
| top-5 programs | .299 | .516 | 184.16 | .652 | .554 | 153.28 | .848 | .829 | 30.59 | .283 | .277 | 62.04 |
| FCFU | .456 | **.813** | 489.53 | **.889** | **.853** | 606.65 | - | - | > 1440 | **.338** | **.339** | 319.60 |
| dPads | **.458** | .812 | 147.87 | .887 | **.853** | 348.25 | **.945** | **.939** | 174.68 | .337 | .337 | 162.70 |

available architectures. These partitions are pushed back to the queue after fine-tunning (similar to dPads). Once a graph with each node containing at most one architecture is obtained from the queue as the least cost, we immediately unfold the graph into deeper levels and push it back to the queue unless the maximum depth is reached. FCFU prioritizes to unfold the best partial program observed so far. We compared dPads with FCFU and top-$N$ programs over 5 random runs. The results are shown in Table 15.

It can be seen that dPads outperforms *top-N programs* where $N$ is set to 5, achieving higher accuracy and $F_1$ scores. This is because *top-N programs* tends to excessively detach many valid DSL functions from graph nodes (especially when $N$ is small), leading to suboptimal final programs. FCFU achieves comparable accuracy and $F_1$ scores with dPads but runs significantly slower than dPads. It even times-out on the Basketball dataset. This is because FCFU only unfolds one best partial program each time, causing the search queue to grow exponentially longer with graph decomposition. In contrast, dPads maintains a set of high-quality programs via top-N preservation on nodes and simultaneously expands all of these programs via iterative graph unfolding.

## C.5 Program Examples

We show more synthesized programs by dPads for the four datasets below.

**Crim13.** The following is a programmatic classifier learned for Crim13.

```
Map(
    if AccelerationAffine_{θ_1}(x_t) ≥ 0
    then if PositionAffine_{θ_2}(x_t) ≥ 0
        then PositionAffine_{θ_3}(x_t)
        else VelocityAffine_{θ_4}(x_t)
    else Multiply(DistanceAffine_{θ_5}(x_t), DistanceAffine_{θ_6}(x_t)))  x
```

In the program, AccelerationAffine, DistanceAffine, PositionAffine, and VelocityAffine are functions that first select the parts of the input that represent acceleration, distance, position, and velocity measurements, respectively, and then apply affine transformations with parameters $\theta$ to the resulting

vectors. The program contains a nested if-then-else (ITE) operation conditioned on the acceleration of two mice. Our interpretation of the program is that if the difference between the accelerations of two mice is small, in the first else branch, they are doing "sniff" if the two mice are close to each other without obvious movements (i.e. the multiplication of their distance is small); otherwise, in the first then branch, the program evaluates the likelihood of "sniff" by applying a position bias, then using the velocity of the mice if the mice are close together and not moving fast, and using distance between the mice otherwise.

We show another programmatic classifier learned for Crim13 below:

```
MapPrefixes(
    Fold(
        if DistanceAffine_{θ_1}(x_t) ≥ 0
        then PositionAffine_{θ_2}(x_t)
        else PositionAffine_{θ_3}(x_t)))  x
```

This program has a simpler architecture compared with the one above but has a higher $F_1$ score (0.468 vs. 0.456) that is comparable to the RNN baseline (vs. 0.481). It exploits the distance and the position bias between two mice to evaluate if they are doing "sniff".

**Fly-vs-fly.** We draw two programmatic classifiers for Fly-vs-fly with similar structures below. Both

```
Fold(                                Fold(
  Add(                                 Add(
    Add(PositionalAffine_{θ_1}(x_t),       Add(AngularAffine_{θ_1}(x_t),
        WingAffine_{θ_2}(x_t)),                 WingAffine_{θ_2}(x_t)),
    RatioAffine_{θ_3}(x_t)))  x          Add(RatioAffine_{θ_3}(x_t),
                                                 RatioAffine_{θ_4}(x_t))))  x
```

programs consider the wing and the body ratio features of fruit flies for classification. The left program further extracts the position information and gets a high $F_1$ score 0.904 while the right program extracts the fly angle velocity and relative facing angle information that results in a lower $F_1$ score 0.888. The better performance of the left program suggests that the position information of flies is more crucial than the angle features to classify their actions.

**SK152.** For SK152, we show a learned programmatic classifier that is more complex compared with the one depicted in the main paper.

```
SlideWindowAvg(
  Add(
    if ArmsXYZAffine_{θ_1}(x_t) ≥ 0
    then ArmsXYZAffine_{θ_2}(x_t) else ArmsXYZAffine_{θ_3}(x_t),
    if LegsXYZAffine_{θ_4}(x_t) ≥ 0
    then FaceXYZAffine_{θ_5}(x_t) else LegsXYZAffine_{θ_6}(x_t)))  x
```

This program applies a sliding window to a trajectory for classification and sums the results of two ITE operations inside the window. The first ITE operation focuses on human arm behaviors and the second ITE operation leverages either face movements or leg movements based on the behavior of human legs. The program results in an $F_1$ score of 0.336 and an accuracy of 0.337.

**Basketball.** The program synthesized for Basketball to classify the ballhandler is shown below.

```
Map(
  Multiply(
    Add(BallXYAffine_{θ_1}(x_t), OffenseXYAffine_{θ_2}(x_t)),
    Add(OffenseXYAffine_{θ_3}(x_t), BallXYAffine_{θ_4}(x_t))))  x
```

In the program, OffenseXYAffineand and BallXYAffine are parameterized affine transformations over the XY-coordinates of the offensive players and the ball. The program structure can be interpreted as computing the distance between the offensive players and the ball to determine the ballhandler. As we aim to recognize the offensive player who holds the basketball, it suffices to only consider the ball

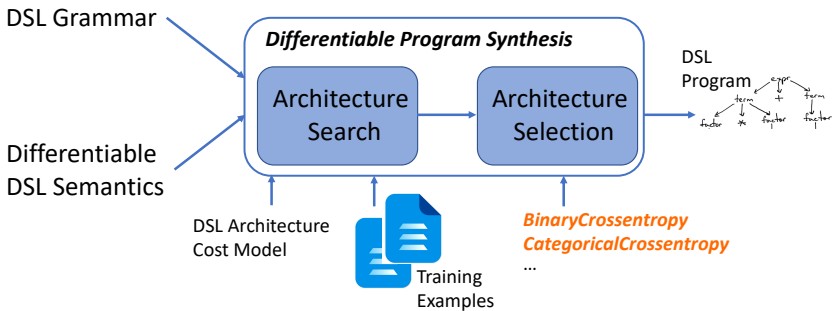

Figure 8: A high-level framework of dPads.

$$\alpha_0 \rightarrow \mathbf{And}(\alpha_1, \alpha_1) \mid \mathbf{Not}(\alpha_1) \mid \mathbf{Or}(\alpha_1, \alpha_1) \mid \mathbf{Xor}(\alpha_1, \alpha_1)$$
$$\alpha_1 \rightarrow \mathbf{And}(\alpha_2, \alpha_2) \mid \mathbf{Not}(\alpha_2) \mid \mathbf{Or}(\alpha_2, \alpha_2) \mid \mathbf{Xor}(\alpha_2, \alpha_2) \mid LN10$$
$$\alpha_2 \rightarrow \mathbf{And}(\alpha_3, \alpha_3) \mid \mathbf{Not}(\alpha_3) \mid \mathbf{Or}(\alpha_3, \alpha_3) \mid \mathbf{Xor}(\alpha_3, \alpha_3)$$
$$\alpha_3 \rightarrow \mathbf{And}(\alpha_4, \alpha_4) \mid \mathbf{Not}(\alpha_4) \mid \mathbf{Or}(\alpha_4, \alpha_4) \mid \mathbf{Xor}(\alpha_4, \alpha_4) \mid LN32$$
$$\alpha_4 \rightarrow LN30 \mid LN42$$

Figure 9: Context-free grammar for cryptographic circuit synthesis in the SyGus project.

positions and the offensive players (excluding the information on defensive players). This program gets a $F_1$ score of 0.945 and an accuracy of 0.939.

## C.6 Generalizability

In the paper, we illustrate dPads by focusing on sequence classification. However, dPads is a general program synthesis algorithm and is not limited to sequence classification.

Fig. 8 depicts a high-level framework of dPads. Other than training data, dPads takes as input a DSL, the semantics of the DSL, and a cost model of each production rule in the DSL. Importantly, dPads requires the DSL's semantics to be differentiable. This is because dPads performs gradient-based architecture search. To avoid discontinuities in programming constructs such as ITE, we require these constructs to be interpreted in terms of a smooth approximation. As such, the synthesis algorithm in dPads is exactly parameterized by a class of DSLs with differentiable semantics. The context-free grammar in Fig. 1 is such an example. It is straightforward to apply dPads to another DSL with differentiable semantics.

As an example, we demonstrate the generalizability of dPads by applying it to the cryptographic circuit synthesis task in the SyGuS (Syntax-Guided Synthesis) project[1]. The goal is to synthesize a side-channel free cryptographic circuit by following given context-free grammar while ensuring that the synthesized circuit is equivalent to the original circuit (a correctness constraint). The grammar is designed to avoid side-channel attacks, whereas the original circuit is created only for functional correctness and thus is vulnerable to such attacks. dPads takes as input a circuit grammar as depicted in Figure. 9. The grammar includes several Boolean operations **And**, **Not**, **Or** and **Xor**. It also specifies *multiple variables* (e.g. $LN10$ and $LN30$) to be used by the synthesizer to generate a desired program (circuit in this context). In this experiment, we aim to synthesize a program that must be logically equivalent to $\varphi_{spec}$ (a correctness specification):

$$\varphi_{spec} : (((LN30 \; \mathbf{Xor} \; LN32) \; \mathbf{Xor} \; LN42) \; \mathbf{Xor} \; LN10) \tag{6}$$

Notice that the above program itself cannot be expressed using the above grammar.

In order to apply dPads to such a task, the user of dPads must provide a differentiable DSL semantics. Recall that we have provided a smooth approximation of ITE in Sec.2. We define a differentiable

---

[1]https://sygus.org/

$$[\![\mathbf{And}(\alpha_1, \alpha_2)]\!](v) = \mathbf{min}([\![\alpha_1]\!](v), [\![\alpha_2]\!](v))$$
$$[\![\mathbf{Or}(\alpha_1, \alpha_2)]\!](v) = \mathbf{max}([\![\alpha_1]\!](v), [\![\alpha_2]\!](v))$$
$$[\![\mathbf{Xor}(\alpha_1, \alpha_2)]\!](v) = [\![\alpha_1]\!](v) + [\![\alpha_2]\!](v) - 2[\![\alpha_1]\!](v) \cdot [\![\alpha_2]\!](v)$$
$$[\![\mathbf{Not}(\alpha)]\!](v) = 1 - [\![\alpha]\!](v)$$
$$[\![LN10]\!](v) = v[LN10]$$
$$[\![LN30]\!](v) = v[LN30]$$
$$[\![LN32]\!](v) = v[LN32]$$
$$[\![LN42]\!](v) = v[LN42]$$

Figure 10: Differentiable semantics for Boolean operations.

semantics for **And**, **Not**, **Or** and **Xor** similarly in Figure 10. Given a program $\varphi$ in the DSL and a Boolean assignment $v$ as the variables in $\varphi$, $\varphi(v)$ is deemed to be True if $[\![\varphi(v)]\!]$ is closer to 1 and $\varphi(v)$ is deemed to be False if $[\![\varphi(v)]\!]$ is closer to 0.

dPads constructs a program derivation graph to include each possible program allowed by the grammar. Given a set of input-output examples, it trains the architecture weights of the program derivation graph by minimizing the MSE loss between the graph's outputs (Equation. 3) and the ground truth outputs. In this example, an input is an assignment to the variables, e.g. $v = \{LN10 : 0,\ LN30 : 1,\ LN32 : 0,\ LN42 : 1\}$. The corresponding output is whether the input variable assignment $v$ should be evaluated to 1 (True) or 0 (False). We collect the input-output examples using counterexample-guided inductive synthesis (CEGIS) by iteratively querying an SMT solver (such as Z3[2]) whether a synthesized program $\varphi$ is logically equivalent to $\varphi_{spec}$. Any counterexample that witnesses the inequivalence of $\varphi$ and $\varphi_{spec}$ is added to the input-output example set. Since the DSL semantics is differentiable, dPads can efficiently learn architecture weights using gradient descent optimization and hence return the best program it synthesizes. For our example, dPads synthesizes the following solution that is verified equivalent to $\varphi_{spec}$:

$$(LN10\ \mathbf{Xor}\ ((LN32\ \mathbf{Xor}\ (LN42\ \mathbf{Or}\ LN30))\ \mathbf{Xor}\ ((LN30\ \mathbf{Or}\ LN42)\ \mathbf{And}\ (LN30\ \mathbf{And}\ LN42))))$$

In our experience, dPads is very efficient in solving the circuit synthesis problem and can reduce the synthesis time from minutes by EUSolver[3] (an enumerative SyGuS solver) to seconds.

# D Additional Discussions

## D.1 Limitations

**Performance gap to RNN.** dPads's performance does not match the RNN baseline. This is mainly due to the limitations in expressivity imposed by the DSL. Firstly, in the DSL, we only allow for customized feature functions $F_{S,\theta}(x)$ that extract a vector consisting of a predefined subset $S$ of the dimensions of an input sequence $x$ and pass the extracted vector through a linear function with trainable parameters $\theta$. For example, for Crim13, we predefine feature functions such as the XY positions, angles, velocity, acceleration, distance of a pair of mice, and distance difference for every two consecutive frames. We list the details of $F_{S,\theta}(x)$ for each dataset in Appendix B. These feature functions are extremely helpful to ensure that a synthesized program composed of these functions is interpretable. However, a program limited to these predefined feature functions may be suboptimal as only a subset of features is used. Instead, an RNN policy can understand the whole context of a sequence using all features available from the input. Secondly, for the sake of interpretation, our DSL also predefines a limited set of algebraic operations to process the outputs from the feature functions, as shown in Fig. 1. However, an RNN can use a more expressive activation function to learn about long-term dependencies in data. We have reported the performance limitation of dPads in Table 1.

**Program Derivation Graph Accuracy.** Another important limitation is that performance estimation of each program included in a program derivation graph ranked by architecture weights can be

---

[2]https://github.com/Z3Prover/z3
[3]https://bitbucket.org/abhishekudupa/eusolver/src/master/

inaccurate due to the co-adaption among partial architectures (node sharing). As one super program derivation graph may not be able to model the entire search space accurately, we have used multiple sub program derivation graphs generated on the fly via a search procedure to address this problem (Section 3.3). Each sub derivation graph models one part of the search space. However, as reported in Table. 3, the search method slows down the whole synthesis procedure and may need additional optimization.

## D.2 Further Comparison with NEAR

Although dPads and NEAR [7] both formalize program synthesis as a graph search problem, the two techniques are very different. We compare dPads and NEAR in depth as follows.

**NEAR.** Typically, search-based program synthesizers enumerate the underlying program space in some order and for each program checks whether or not it satisfies the synthesis constraints. It is a challenging problem because the architecture search space is combinatorial. The most simple strategy that starts by searching for programs with 1 DSL production rule and iteratively increases this bound does not scale to complex programs. NEAR uses neural models to approximate unexpanded subexpressions to estimate the likelihood of eventually deriving a high-quality program by choosing a particular production rule. It leverages this kind of information to prioritize promising search directions and hence can greatly accelerate the search process. However, NEAR's search strategy is still discrete and enumeration-based.

**dPads's Contributions.** dPads proposes a new, scalable program synthesis technique. It views program architecture search as learning a probability distribution over all possible program architectures induced by a DSL. Unlike NEAR that enumerates and evaluates each program by training unknown parameters from scratch, dPads reduces the computation cost by training one program, a.k.a. a program derivation graph, to approximate the performance of every program in the search space. It learns the architecture weights of each program encoded in a program derivation graph in a way that ranks the performance estimation of these programs. The search procedure in dPads is more efficient because it supports gradient-based architecture optimization in a novel continuous relaxation of the program architecture search space.

**dPads's Impacts.** To the best of our knowledge, dPads is the first program synthesis technique that applies gradient-based architecture search. Experiment results on sequence classification demonstrate that thanks to this strategy, dPads can efficiently search in a deep program space to learn sophisticated programs that are necessary to ensure high $F_1$ scores.

## D.3 Using Neural Networks to Approximate Missing Expressions

Both dPads and NEAR [7] use neural networks to approximate missing expressions of partial architectures. However, the key difference is that dPads does not use neural models to estimate the performance of a partially expanded program. This is because our experiments find that due to overfitting or underfitting, a neural model may be biased on a particular program. In contrast, dPads uses a sub program derivation graph to estimate the performance of a partially expanded program. We find that it can provide more accurate assessment due to the graph's syntax resemblance to a valid program. This enables dPads to be able to search deeper than NEAR in the program architecture space to synthesize more sophisticated programs that are necessary to ensure higher $F_1$ scores. For example, on Fly-vs-fly, NEAR's inaccurate admissible neural heuristics prevent it from searching the architecture space deeper than that contains programs derived by just 2.8 production rules. In contrast, during architecture selection, dPads uses a sub program derivation graph to estimate the performance of a partially expanded program. The sub program derivation graph provides a more accurate assessment. As a result, dPads searches the architecture space much deeper containing programs derived by 6.8 production rules and gets a much higher $F_1$ score (.887 vs .828).

dPads only uses neural models to provide contrastive performance estimation for a set of programs sharing nodes in a program derivation graph for iterative graph unfolding (Section 3.2). For example, on the Basketball dataset, the architecture spaces searched by the two tools are roughly equivalent. dPads is 3 times faster. This is in part because dPads uses the same set of neural networks to provide contrastive performance estimation for the set of programs sharing nodes and does require the performance estimation on a single program to be accurate. In iterative graph unfolding, this strategy effectively and efficiently prunes away a large set of unlikely search directions.