# OpenReview forum: "Differentiable Synthesis of Program Architectures"
_NeurIPS.cc/2021/Conference — NeurIPS 2021 Poster_

### Official Review · Reviewer_Rykf · 2021-07-15

**Rating:** 6
**Confidence:** 2

**Summary:**

A method for synthesising programs using gradient descent is presented, alternating between optimisation of operator parameters (constants) and of the program structure (nested expressions). The submission focuses on reducing the complexity of the search for correct structures, using the idea of sharing different nodes in the search tree (i.e., allow them to appear in different contexts) as well as initially approximating subexpressions by neural networks and only starting to learn to represent them by programs when the subtree appears to be a promising candidate.
Experiments on four small datasets show that the method compares favourably to a baseline in synthesizing programs that explain observed behaviour.

**Limitations And Societal Impact:**

In the checklist, the authors claim that they discussed the limitations of their work, but I could find no such discussion in the paper. No negative societal impact is apparent.

**Main Review:**

Originality
---
I'm not an expert in this area, and hence cannot comment much on this dimension. However, the idea of re-using parts of the synthesized programs in several places was already present in the TerpreT work of Gaunt et al, which considered a number of formalisms (e.g., assembly programs) in which operand choices were shared in a similar way as in the proposed "Node Sharing" method in this submission.

Quality
---
The only claims in the paper are related to the effectiveness of the proposed method. Claims regarding the overall method seem well-supported by the experiments (but again, not an expert and hence no idea if these datasets/baselines are appropriate), but the individual contribution of the different methods remain unclear:
1. The experiments do not break out the effect of the node sharing technique at all. For example, what would happen if every potential function would have its own child nodes? Clearly, the runtime would explode, but would the quality of the generated programs improve as well? You could even imagine faster convergence, as the number of constraints on each parameter is reduced.
2. The effect of the iterative search tree deepening technique is not studied in detail. In particular, I'm not sure if N=3 means that all children are always preserved (that somewhat depends on how the $c$ parameter of fold is treated); and many other pruning approaches (e.g., considering the top N candidates across the entire search tree, rather than per node) are not considered at all.

Clarity
---
I found the paper very hard to follow, in particular because some core concepts are not illustrated in a helpful way. For example, it would have been helpful to explain the meaning of $w$ - I first thought this is simply the probability distribution over potential choices in a node; however this seems to be wrong when considering Fig. 3 and Eq. (3), and instead it turns out that this is conditional on where it is used. Providing a full example would have made following the text much easier.

There are a number of imprecisions (or maybe missed opportunities):
* l162-l165: What about the first parameter of ITE and Add? Are these not shared (because this wouldn't be well-typed), or is there some special-casing happening here? More generally, are types used at all to direct the search?
* how are variables picked by the system? does the grammar contain a fixed pool of variables, or is this somehow special-cased?
* Sect. 3.3: the choice of the name $f$ to compute a cost in Alg. 1 is extremely unfortunate, as it has no relationship to the $f_u^k$ also appearing here.

Significance
---
The present submission does not seem to be a breakthrough in applicability of neural program search to complex problems. The idea of iterative deepening of the search is interesting, though somewhat underexplored in this submission, and just aims to bound the base of the exponential blowup of the search method.


Minor details
---
- Fig. 1: I found this DSL extremely strange (without knowing the datasets), as it mixes non-standard, apparently task-focused primitives such as mapprefix and SlideWindowAvg into a basic functional language. It would be helpful to comment on the choice of functions in the text.
- l105: "sum of the multiset of rules" -> "sum of the costs of the rules in the multiset of rules" (or something slightly less ungainly, but correct)
- l135: "are inefficient in natural" -> "... in general"
- l204: "noterminals"
- l223: "depcited"
- Fig. 6: left half of table for "Sk152-10 actions" is a copy of the Crim13-sniff results

Update after Author Response
---
Given the additional example on a different DSL, I'm happy to believe that this submission is more significant as it is applicable to a wider range of examples; hence I'm upgrading my original rating.

**Time Spent Reviewing:**

4

---

> ### Author Response · Authors · 2021-08-10
> **Review Response**
>
> We appreciate your effort in providing detailed and helpful reviews. We address the concerns and questions as follows:
>
> ## Q1: Not a breakthrough in applying neural program search to complex problems.
>
> Broadly speaking, dPads is different than neural program induction techniques, as the outputs of these approaches are neural networks as opposed to programs. dPads is also different than metalearning-based approaches such as [21, 34] because dPads must learn a program from a single dataset. We believe learning from a single dataset is an important research problem because in many applications we may not have a corpus of datasets and corresponding programs available.
>
> State-of-the-art program learning methods on learning from a single dataset such as NEAR typically enumerate the underlying program space in some order and for each program checks whether or not it satisfies the synthesis constraints by training unknown program parameters from scratch. Since the search space can be intractably large, enumeration-based strategies are inefficient in general.
>
> dPads is the *first* program synthesis algorithm that applies gradient-based search in a program architecture space. It efficiently reduces the computation cost of enumeration by training one program, a.k.a. a program derivation graph, to approximate the quality (e.g. classification accuracy) of every program in the search space. It learns the architecture weights of each program encoded in a program derivation graph in a way that ranks the performance estimation of these programs. Experiment results demonstrate that thanks to this strategy, dPads can efficiently search in a much deeper program space than enumeration-based methods such as NEAR to learn more complex programs that are necessary to ensure higher F1 scores. NEAR often cannot reach the necessarily deep program space within the time budget because of its less efficient enumeration-based search strategy and less effective neural heuristics (see Table 8 in the appendix for more details).
>
>
> ## Q2: Node-sharing not novel or similar to the TerpreT work of Gaunt et al.
>
> We agree with the reviewer that node sharing itself is not novel. Other than TerpreT, many neural architecture search methods heavily use this idea to accelerate search. However, dPads's core contribution is the *first* program synthesis algorithm that applies gradient-based search in the discrete space of program architectures.
>
> More specifically, existing differentiable programming techniques including TerpreT consider a parameterized representation of programs (essentially, an "architecture" in the language of our paper), and the learning objective is to find optimal parameters for this representation. In contrast, dPads searches over the nonparametric space of all architectures expressed in a rich programming language. The only methods we know that searches over architectures of differentiable programs are based on search space enumeration, e.g. NEAR. For the first time, dPads demonstrates that gradient-based architecture search is applicable to program synthesis and it substantially outperforms enumeration-based program learning methods. Node sharing and iterative graph unfolding are two optimizations that further scale dPads to complex programs.
>
> ## Q3: What would happen if every potential function would have its own child nodes? What is the effect of the iterative search tree deepening technique?
>
> We started our research without considering node sharing and iterative graph unfolding but found that dPads does not work well without these two optimization strategies. The reason is that the size of a program derivation graph grows exponentially large with the height of its search tree (e.g. Figure 2) and the number of DSL functions (e.g. Fold and ITE). Without the two optimizations, limited by the size of GPU memory, dPads may either time-out or encounter out-of-memory error when searching programs that need deep structures to ensure high accuracy. We report the comparison results over 5 random runs as follows. Costs of time are set in minutes.
>
> |                  | Crim13 |      |        | Fly-vs-fly |      |        | Bball |      |        | Sk152 |      |        |
> |:----------------|:------:|:----:|:------:|:----------:|:----:|:------:|:-----:|:----:|:------:|:-----:|:----:|--------|
> |                  | **F1**     | **Acc.** | **Time**   | **F1**         | **Acc.** | **Time**   | **F1**    | **Acc.** | **Time**   | **F1**    | **Acc.** | **Time**   |
> | **dPads w/o node sharing**  | .453   | .800 | 334.93 | -          | -    | >24hrs | -     | -    | >24hrs | .321  | .322 | 252.81 |
> | **dPads w/o graph unfolding** | .449   | .818 | 280.67 | -          | -    | OOM    | .848  | .832 | 348.09 | .348  | .346 | 273.95 |
> | **dPads**            | .458   | .812 | 147.87 | .887       | .853 | 348.25 | .945  | .939 | 174.68 | .337  | .337 | 162.70 |
>
>
> The results demonstrate that dPads with the two optimization strategies converges much faster.
>
> On Fly-vs-fly, without iterative graph unfolding, the program derivation graph generated by dPads has 33 nodes shared by 485 operations, 32 edges, and a total of 13824 parameters to train. dPads encounters an out-of-memory (OOM) exception when training this graph. In contrast, with iterative graph unfolding, the deepest graph has only 21 nodes shared by 220 operations, 20 edges, and a total of 5497 parameters to train. dPads converges in less than 360 mins.
>
> On Basketball, without node sharing, the deepest program derivation graph generated by dPads has 94 nodes hosting 302 operations, 93 edges, and a total of 8637 parameters to train. The search procedure of dPads (Sec 3.3) does not terminate in 24 hours because of the many nodes and edges. In contrast, with node sharing, the program derivation graph reduces to only 29 nodes shared by 152 operations, 28 edges, and a total of 4157 parameters. dPads converges in less than 180 mins.
>
> Moreover, training without these two optimizations does not necessarily produce better results even when there is not OOM or timeout. For example, on Basketball, dPads achieves .945 F1 score. dPads without iterative graph unfolding only obtains .848 F1 score. We suspect this is because the program derivation graph without iterative unfolding is more difficult to train as it contains significantly more parameters (6791 vs 4157). In contrast, dPads uses top-N preservation to efficiently prune away low-quality options. We will provide additional details in the final version.
>
>
> ## Q4: Lack of study on many other pruning approaches (e.g., considering the top-N candidates across the entire search tree, rather than per node)?
>
> We indeed have explored other pruning approaches for iterative graph unfolding, including reserving the top-N candidates across the entire search graph rather than per node, which we refer to as top-N programs. We also considered another pruning algorithm which we refer to as First Complete First Unfold (FCFU). It leverages A$^*$ search to manage a queue of program derivation graphs. After training converges on a program derivation graph from the queue, FCFU decomposes the graph into several sub graphs which are pushed back to the queue. Once a graph with each node containing at most 1 DSL function is obtained from the queue as the least cost, we immediately unfold the graph into deeper levels and push it back to the queue. FCFU prioritizes to unfold the best partial program observed so far.
>
> We compared dPads with FCFU and top-N programs (as suggested by the reviewer) over 5 random runs. The results are as follows. Costs of time are set in minutes.
>
> |            | Cim13 |      |        | Fly-vs-fly |      |        | Bball |      |        | Sk152 |      |        |
> |------------|-------|------|--------|------------|------|--------|-------|------|--------|-------|------|--------|
> |            | **F1**    | **Acc.** | **Time**   | **F1**         | **Acc.** | **Time**   | **F1**    | **Acc.** | **Time**   | **F1**    | **Acc.** | **Time**  |
> | **FCFU** | .456  | .813 | 489.53 | .889       | .853 | 606.65 | -     | -    | >24hrs | .338  | .339 | 319.60 |
> | **top-5 programs** | .299  | .516 | 184.16 | .652       | .554 | 153.28 | .848  | .829 | 30.59  | .283  | .277 | 62.04  |
> | **dPads**      | .458  | .812 | 147.87 | .887       | .853 | 348.25 | .945  | .939 | 174.68 | .337  | .337 | 162.70 |
>
> It can be seen that dPads outperforms top-5 programs preservation, achieving higher accuracy and F1 scores. This is because top-N programs preservation tends to excessively detach many valid DSL functions from graph nodes (especially when N is small), leading to suboptimal final programs. FCFU achieves comparable accuracy and F1 scores with dPads but runs significantly slower than dPads. It even times-out on Basketball. This is because FCFU only unfolds one best partial program each time, causing the search queue to grow exponentially longer with graph decomposition. In contrast, dPads maintains a *set* of high-quality programs via top-N preservation on nodes and simultaneously expands all of these programs via iterative graph unfolding. We will be happy to provide additional details in the final version if the paper is accepted.

---

> > ### Author Response · Authors · 2021-08-10
> > **Review Response (Continue)**
> >
> > ## Q5: No discussion of limitation.
> >
> > As pointed out by the other reviewers, our paper acknowledges that the main technical limitation is that dPads's performance does not match the RNN baseline. This is mainly due to the limitations in expressivity imposed by the DSL. Firstly, in dPads's DSLs, we only allow for customized feature functions $F_{S,\theta}(x)$ that extract a vector consisting of a predefined subset $S$ of the dimensions of an input sequence $x$. For example, for Crim13, we predefine feature functions such as the XY positions, angles, velocity, acceleration, distance of a pair of mice, and distance difference for every two consecutive frames. We list the details of $F_{S,\theta}(x)$ for each dataset in Table 3, 4, 5, 6 in Appendix C. These feature functions are extremely helpful to ensure that a synthesized program composed of these feature functions is interpretable. However, a program limited to these predefined feature functions may be suboptimal as only a subset of features is used. Instead, an RNN policy can understand the whole context of a sequence using all features available from the input. Secondly, for the sake of interpretation, the DSLs also predefine a limited set of algebraic operations to process the outputs from the feature functions. However, an RNN can use a more expressive activation function to learn about long-term dependencies in data. We have reported the performance limitation of dPads in Table 1.
> >
> > We also mention in Section 6 that there is potential for attacks on security and fairness on programmatic models derived via dPads.
> >
> > ## Q6: Can the first parameter of ITE and Add share parameters? More generally, are types used at all to direct the search?
> >
> > The ITE function is an abbreviation for "if $\alpha_1$ $\ge$ 0 then $\alpha_2$ else $\alpha_3$". Therefore $\alpha_1$ does not have a Boolean type and we allow for nested ITEs.
> >
> > Programs in our DSLs operate over two data types: real vectors and sequences of real vectors. We assume a simple type system that makes sure that these types are used consistently. The type system is quite basic and primarily ensures that the types of formal parameters, actual parameters, and return values are as expected. When expanding a partial architecture, we ensure that the chosen expansion is consistent w.r.t. this type system. We will provide additional details about the type system in the final version.
> >
> > ## Q7: DSL is extremely strange, as it mixes non-standard, apparently task-focused primitives.
> >
> > The programming language in Figure 1 is a domain-specific language (DSL), specific to the task of sequence classification. Although dPads is generally applicable in many applications, we specifically study it in the sequence classification context. Thus, we sketch our DSL for this domain in the paper. For example, the DSL includes a set of higher-order combinators to recurse over sequences that aim to compactly express sequence-to-sequence functions.
> >
> > The way that we define our DSLs is standard in research on Programming Languages (PL), and also appears in many prior papers in the intersection of PL and Machine Learning.
> >
> > ## Q8: How are variables picked? Does the grammar contain a fixed pool of variables?
> >
> > Like many other DSLs for program synthesis, the programming language in Figure 1 is purely functional. Thus it only needs to deal with the input sequence $x$. We also allow for customized feature functions $F_{S,\theta}(x)$ that extract a vector consisting of a predefined subset $S$ of the dimensions of a sequence $x$ and pass the extracted vector through a linear function with trainable parameters $\theta$.
> >
> > ## Q9: Does N=3 mean that all children are always preserved?
> >
> > N=3 does not mean that we preserve all children. The program derivation graph in Figure 3 is only an example. In our experiment, a node typically contains more than 3 operations as our DSLs contain a lot more than 3 functions. We give the full details of our DSLs in Appendix C.
> >
> > ## Q10: Left half of Table 2 for "Sk152-10 actions" is a copy of the Crim13-sniff results.
> >
> > We apology for the confusion. We have corrected Table 2 in Appendix A.
> >
> > **Other Comments.** In addition to these main points, we want to thank the reviewer for the helpful suggestions on how to further improve the paper. We plan on incorporating all of these, such as exactly defining the symbols $w$ and $f$, commenting on the choice of DSL functions, and discussing the limitations of dPads in depth in the final version.

---

> > > ### Comment · Reviewer_Rykf · 2021-08-11
> > > **Response Discussion**
> > >
> > > > The ITE function is an abbreviation for "if $\alpha_1 \geq 0$ then $\alpha_2$ else $\alpha_3$". Therefore does not have a Boolean type and we allow for nested ITEs.
> > >
> > > If that is the case, please fix Fig. 1 and l93/l94 - your provided semantics treats $\alpha_1$ as a full subexpression. This explains quite a bit of confusion I had during reading of your paper.
> > >
> > > > The programming language in Figure 1 is a domain-specific language (DSL), specific to the task of sequence classification.
> > > > Although dPads is generally applicable in many applications, we specifically study it in the sequence classification context.
> > >
> > > This is the part I'm struggling with - quite a few of your modelling choices seem to be specific to the DSL for this example:
> > > * supporting a single variable (which would break in the presence of several inputs)
> > > * typing isn't required because your DSL only supports a very limited set of values
> > > * no conditionals beyond $\geq 0$
> > > It is not directly clear how to adapt dPad to less restricted DSL, and I find that concerning.
> > >
> > > > The way that we define our DSLs is standard in research on Programming Languages (PL), and also appears in many prior
> > > > papers in the intersection of PL and Machine Learning.
> > >
> > > PL papers usually present methods that are parametrized by a class of DSLs, whereas here, the limitations of the picked DSL seems to be crucially used to define the model. What concerns me, concretely, is overfitting the method to a small set of (let's all be honest here: practically entirely irrelevant) benchmarks, rather than making progress on a general methods. I'm trying to understand if that is the case or not.
> > >
> > > Additionally, `fold` (together with a list or tuple constructor) would be entirely sufficient to express the other higher-order functions in your DSL; `SlideWindowAvg` is _not_ a standard function, but one that is carefully selected to be a useful primitive for the four benchmarks you are considering - that is the part I'm chafing at, but I understand this DSL has been picked by authors before you (in particular, Shah et al define a DSL without the sldiing window average, and then slip that back in as a side comment in the experiment section...)

---

> > > > ### Author Response · Authors · 2021-08-12
> > > > **DSL Discussion: Applying dPads to a new DSL**
> > > >
> > > > We thank the reviewer for the valuable comments!
> > > >
> > > > > What concerns me, concretely, is overfitting the method to a small set of benchmarks, rather than making progress on a general method.
> > > >
> > > > This is not the case. In the paper, we illustrate dPads by focusing on sequence classification. However, dPads is a general program synthesis algorithm and is not limited to sequence classification.
> > > >
> > > > Other than training data, dPads takes as input *a DSL* and *the semantics of the DSL*. Importantly, dPads requires the DSL's semantics to be smooth and support differentiation. This is because dPads performs gradient-based architecture search. To avoid discontinuities in programming constructs such as ITE, we require these constructs to be interpreted in terms of a smooth approximation. In this review response, we will provide a few examples of how common non-smooth constructs can be smoothly approximated.
> > > >
> > > > As such, the synthesis algorithm in dPads is exactly parameterized by a class of DSLs with smooth and differentiable semantics. The context-free grammar in Figure 1 of the paper is such an example. It is straightforward to apply dPads to another DSL with smooth and differentiable semantics. Please see an end-to-end example of applying dPads to a new DSL below.
> > > >
> > > > >  Quite a few of your modelling choices seem to be specific to the DSL for this example
> > > >
> > > > **Single variable.** In the paper, we focus on sequance classification that takes only a sequence of real vectors as input. Our paper provides a simplified single-input version of the full semantics because we intended to avoid confusion. dPads actually generalizes to programs that take multiple inputs. In fact, we assume that a program takes as input a map $v$ that is from input variables to values. For example, if a program takes two input parameters $x_1$ and $x_2$. An input to the program is a map $v = \\{ x_1: \nu_1, x_2: \nu_2 \\}$ where $\nu_1 = v[x_1]$ (or $\nu_2 = v[x_2] $) is the value of $x_1$ (or $x_2$). We provide a concrete synthesis example for multiple variables below.
> > > >
> > > > The formal semantics we use in dPads is as follows where we use $[[\alpha]]$ to encode the semantics of a syntax construct $\alpha$:
> > > >
> > > > $[[x]] (v) = v[x]$ where $v[x]$ is the value of the variable $x$ in an input $v$.
> > > >
> > > > $[[c]] (v) = c$ where $c$ is a constant.
> > > >
> > > > $[[\textbf{ADD}(\alpha_1,\alpha_2)]] (v) =[[\alpha_1]] (v)+[[\alpha_2]] (v) $
> > > >
> > > > $[[\textbf{Multipy}(\alpha_1,\alpha_2)]] (v) =[[\alpha_1]] (v) \cdot [[\alpha_2]] (v) $
> > > >
> > > > $[[\textbf{ITE}(\alpha_1,\alpha_2,\alpha_3)]] (v) = \sigma([[\alpha_1]]  (v)) \cdot [[\alpha_2]] (v) + (1-\sigma([[\alpha_1]] (v))) \cdot [[\alpha_3]] (v)$
> > > >
> > > > $[[\textbf{map}\ (\textbf{fun}\ x_1. \alpha_1 )\ x]] (v) = \Big[ [[\alpha_1]] (v_0),\ \ldots,\ [[\alpha_1]] (v_n) \Big]$ where $v_j$ is the $j$-th element in $[[x]] (v)$
> > > >
> > > > We elide the semantics of the other DSL constructs. We will add the formal semantics definition to the final version of the paper.
> > > >
> > > > Importantly, our semantics is differentiable because we have provided a smooth approximation of the semantics of $\text{ITE} (\alpha_1, \alpha_2, \alpha_3)\ \equiv\ \text{if} (\alpha_1 \ge 0)\ \text{then}\ \alpha_2\ \text{else}\ \alpha_3$ above where $\sigma$ is the sigmoid function. Notice that the type of $\alpha_1$ is not Boolean. The intuition is that if $[[\alpha_1]]  (v)$ is signicantly greater than 0, $\sigma([[\alpha_1]]  (v))$ is closer to 1 and the then branch $\alpha_2$ would dominate in the output. Conversely, if $[[\alpha_1]]  (v)$ is signicantly less than 0, the else branch $\alpha_3$ would dominiate in the output. Based on this discussion, it seems that our explanation of the semantics of ITE in the paper is correct. We appreicate it if the reviewer could provide additional clarification.
> > > >
> > > > **Typing.** A type system is needed in dPads to eliminate ill-typed programs from a program derivation graph. Even for sequence classification, it is needed. For example, we type check that any input to and output from Map must be a sequence of vectors and a linear function should only handle a vector.
> > > >
> > > > **No conditionals beyond $\ge 0$.** Although dPads can use nested ITEs to express programs that need arbitrary Boolean combinations to influence control flows, dPads also directly supports arbitrary Boolean combinations in a DSL. In the following example (already implemented), we show how dPads can be used to synthesize programs in a DSL with both **multiple variables** and **arbitrary Boolean combinations**.
> > > >
> > > > # Applying dPads to a new DSL (an end-to-end example)
> > > >
> > > > To show that dPads's synthesis algorithm can be applied to any DSLs with differentiable semantics, we apply it to a SyGuS (Syntax-Guided Synthesis https://sygus.org/) problem - cryptographic circuit synthesis [1]. The goal is to synthesize a side-channel free cryptographic circuit by following the given DSL CFG (context-free grammar) while ensuring that the synthesized circuit is equivalent to the original circuit (a correctness constraint). The grammar is designed to avoid side-channel attacks, whereas the original circuit is created only for functional correctness and thus is vulnerable to such attacks.  **We have already applied dPads to synthesize cryptographic circuits**. For example, dPads can take as input a circuit grammar as follows:
> > > >
> > > > $\alpha_0 \rightarrow \textbf{And}\ (\alpha_1,\alpha_1)\ \vert\ \textbf{Not}\ (\alpha_1)\ \vert\ \textbf{Or}\ (\alpha_1,\alpha_1)\ \vert\ \textbf{Xor}\ (\alpha_1,\alpha_1)$
> > > >
> > > > $\alpha_1 \rightarrow \textbf{And}\ (\alpha_2,\alpha_2)\ \vert\ \textbf{Not}\ (\alpha_2)\ \vert\ \textbf{Or}\ (\alpha_2,\alpha_2)\ \vert\ \textbf{Xor}\ (\alpha_2,\alpha_2)\ \vert\ LN10$
> > > >
> > > > $\alpha_2 \rightarrow \textbf{And}\ (\alpha_3,\alpha_3)\ \vert\ \textbf{Not}\ (\alpha_3)\ \vert\ \textbf{Or}\ (\alpha_3,\alpha_3)\ \vert\ \textbf{Xor}\ (\alpha_3,\alpha_3)$
> > > >
> > > > $\alpha_3 \rightarrow \textbf{And}\ (\alpha_4,\alpha_4)\ \vert\ \textbf{Not}\ (\alpha_4)\ \vert\ \textbf{Or}\ (\alpha_4,\alpha_4)\ \vert\ \textbf{Xor}\ (\alpha_4,\alpha_4)\ \vert\ LN32$
> > > >
> > > > $\alpha_4 \rightarrow LN30\ \vert\ LN42$
> > > >
> > > > The grammar includes several Boolean operations **And**, **Not**, **Or**, and **Xor**. It also specifies *mulitple variables* (e.g. LN10 and LN30) to be used by the synthesizer to generate a desired program (i.e. a circuit in this context). For this example, a synthesized program must be logically equivalent to $\varphi_{spec}$ (a correctness specification):
> > > >
> > > > $\varphi_{spec}:\ (((LN30\ \textbf{Xor}\ LN32)\ \textbf{Xor}\ LN42)\ \textbf{Xor}\ LN10)$
> > > >
> > > > Notice that the above program itself cannot be expressed using the above grammar.
> > > >
> > > > The user of dPads must provide a differentiable DSL semantics.  Recall that we have provided a smooth approximation of ITE in the paper and above. We define a smooth and differentiable semantics for **AND**, **Not**, **Or** and **Xor** similarly:
> > > >
> > > > $[[\textbf{AND} (\alpha_1, \alpha_2)]] (v) = \textbf{min}([[\alpha_1]] (v), [[\alpha_2]] (v))$
> > > >
> > > > $[[\textbf{Or} (\alpha_1, \alpha_2)]] (v) = \textbf{max}([[\alpha_1]] (v), [[\alpha_2]] (v))$
> > > >
> > > > $[[\textbf{Xor} (\alpha_1, \alpha_2)]] (v) = [[\alpha_1]] (v) + [[\alpha_2]] (v) - 2 \cdot [[\alpha_1]] (v) \cdot [[\alpha_2]] (v)$
> > > >
> > > > $[[\textbf{Not}(\alpha)]] (v) = 1 - [[\alpha]] (v)$
> > > >
> > > > $[[LN10]] (v) = v[LN10]$, $[[LN30]] (v) = v[LN30]$, $[[LN32]] (v) = v[LN32]$, $[[LN42]] (v) = v[LN42]$
> > > >
> > > > Given a program $\varphi$ and a Boolean assignment $v$ to the variables in $\varphi$, $\varphi(v)$ is likely to be True if $[[\varphi]] (v)$ is closer to 1 and $\varphi(v)$ is likely to be False if $[[\varphi]] (v)$ is closer to 0 (similar to the way of using the sigmoid function to approximate ITE conditional semantics).
> > > >
> > > > dPads constructs a program derivation graph to include each possible program allowed by the grammar (with node sharing). Given a set of input-output examples, it trains the architecture weights of the program derivation graph by minimizing the MSE loss between the graph's outputs (Equation 3 in the paper) and the ground truth outputs. In this example, an input is an assignment to the variables, e.g. $v = \\{ LN10: 0, LN30: 1, LN32: 0, LN42: 1 \\}$. The corresponding output is whether the input variable assignment $v$ should be evaluated to 1 (True) or 0 (False). We collect the input-output examples using counterexample-guided inductive synthesis (CEGIS) by iteratively querying an SMT solver (https://github.com/Z3Prover/z3) if a synthesized program $\varphi$ is equivalent to $\varphi_{spec}$. Any counterexample that witnesses the inequivalence of $\varphi$ and $\varphi_{spec}$ is added to the input-output example set. Since the DSL semantics is smooth and differentiable, dPads can efficiently learn architecture weights using gradient descent optimization and hence return the best program it synthesizes. For our example, dPads synthesizes the following solution that is verified equivalent to $\varphi_{spec}$:
> > > >
> > > > $(LN10\ \textbf{Xor}\ ((LN32\ \textbf{Xor}\ (LN42\ \textbf{Or}\ LN30))\ \textbf{Xor}\ ((LN30\ \textbf{Or}\ LN42)\ \textbf{And}\ (LN30\ \textbf{And}\ LN42))))$
> > > >
> > > > In our experience, dPads is very efficient in solving the circuit synthesis problem and can reduce the synthesis time from minutes by EUSolver (an enumerative SyGuS solver) to seconds.
> > > >
> > > > > fold would be entirely sufficient to express the other higher-order functions
> > > >
> > > > We agree with the reviewer. We use the DSL in Figure 1 because we aim to ensure the comparison to NEAR is fair.
> > > >
> > > > **Revision.** We will clarify in the final version that dPads is a general synthesis algorithm parameterized by a class of DSLs with differentiable semantics. We thank the reviewer for the great suggestion!  We will provide the full semantics of the DSL. We hope that the circuit synthesis example also addresses the reviewer's concern about whether dPads can support multiple variables and conditionals beyond $\ge 0$.
> > > >
> > > > -----------------
> > > > [1] Hassan Eldib, Meng Wu, and Chao Wang. Synthesis of fault-attack countermeasures for cryptographic circuits. Computer Aided Verification, 2016.

---

> > > > > ### Comment · Reviewer_Rykf · 2021-08-12
> > > > > **Response Discussion**
> > > > >
> > > > > Thank you for the very verbose update (by the way, more concise responses would be appreciated, to keep reviewer load low...).
> > > > >
> > > > > The addition of the circuit synthesis is very compelling w.r.t. my concerns re generalizability of the method, and I'm happy update my rating based on this. It would be great if this could be included in an updated version of the paper, at least an appendix with a small reference in the main text.
> > > > >
> > > > > I would also encourage you to update the DSL definition in Fig. 1 to say $ITE(\alpha_1 \geq 0, \alpha_2, \alpha_3)$ or better $if\ \alpha_1 \geq 0\ then\ \alpha_2\ else\ \alpha_3$, to avoid confusion for other readers. (This is similar as to how $map(fun\ x_1.\alpha_1) x$ is spelled out rather than being abbreviated to $map(\alpha_1, x)$)

---

> > > > > > ### Author Response · Authors · 2021-08-12
> > > > > > **Thanks for raising the score!**
> > > > > >
> > > > > > We appreciate the reviewer for raising the score to 6! Thanks for the valuable suggestions!
> > > > > >
> > > > > > As suggested, we will include the circuit synthesis results in the final version of the paper. We will also update the ITE definition in the DSL to avoid confusion for other readers.
> > > > > >
> > > > > > We apology for the long responses. Thanks again for reviewing our paper and responses!

---

### Official Review · Reviewer_AjTJ · 2021-07-16

**Rating:** 5
**Confidence:** 4

**Summary:**

This paper introduces dPads, an approach to learning interpretable programs to perform sequence classification tasks, as an alternative to using RNNs which are less interpretable and possibly less robust. The programs are generated using a context-free grammar (adapted from prior work) of differentiable operations applied to sequences. The ultimate objective is to find a program that obtains high classification accuracy on a dataset, while having low architectural cost. This is done by using a continuous relaxation of the discrete choices in the grammar, using a softmax over all possible grammar production rules instead of a categorical choice, with learnable weights. Further optimizations include “node sharing” (where two nodes in the program derivation graph are shared if they represent the same nonterminal and thus can be expanded the same way) and “iterative graph unfolding” which is somewhat like a beam search over the program derivation process, narrowing down the earlier choices using a guess for their quality, resulting in less combinatorial explosion (higher efficiency) at the cost of a fully complete search. As a final step, an A* search is used to choose a single complete program with high accuracy and low cost. Experiments on 4 sequence classification datasets show that dPads performs better than prior work NEAR, but worse than plain RNNs (which are not interpretable programs).

**Limitations And Societal Impact:**

One point that should be mentioned is that, IIUC, dPads requires more time and resources (i.e., higher carbon footprint) to produce a program, compared to simpler approaches like plain RNNs.

**Main Review:**

**Positives**
* dPads performs better than prior work NEAR
* The diagrams are quite helpful for understanding the approach
* The iterative graph unfolding idea is quite interesting. Even though it’s similar to a beam search, I haven’t seen such an idea presented for a tree-shaped program derivation before.

**Negatives**
* The prose is a bit dense which makes it hard to follow some parts (but the diagrams are quite helpful)
* The work is quite similar to NEAR, with the similarities and differences not adequately explained, which can make many parts of dPads appear more novel than they are
* There is not enough experimental analysis about the impact of the differences between dPads and NEAR

---

Overall the approach in this paper makes sense. It did take me multiple passes through the paper to actually understand the approach, so perhaps the writing could be made more clear, maybe with a higher emphasis on conveying the broader intuitions before diving into details. I found that the diagrams were quite helpful in understanding the approach.

My main concern is that the paper is not clear enough on how dPads differs from NEAR. The two works use very similar DSLs (by design), both view program search as a graph search problem where the graph nodes are partially expanded programs and edges represent grammar production rules, both use A* with admissible heuristics to perform the search within this graph, and both use neural networks to approximate missing (unexpanded) expressions. It appears that dPads implements some engineering optimizations to make the search process faster, but it is not clear whether NEAR uses similar optimizations (since NEAR is faster than dPads on one dataset), what the impact of these optimizations are, nor how novel they are (when viewed at a high level, sharing nodes when nodes represent the same thing is a common technique, and iterative graph unfolding is similar to beam search). Given how similar dPads and NEAR are, I think the paper should have space dedicated to explaining how NEAR works, what its drawbacks are in more detail (maybe with examples), and then demonstrating how dPads’ modifications help address those drawbacks.

As it is, I feel that this paper is a bit misleading because the high similarity to NEAR is swept under the rug, and would only be noticed by a reader already familiar with NEAR. Thus, in my opinion, the paper needs a significant change to present the approach in a way that adequately explains its relationship (similarities and differences and _why_) to prior work. Doing so would also strengthen the paper's significance, if readers can come away with stronger intuitive understandings of why certain approaches are better than others.

**Questions**
* 105: where to do the costs $c(r)$ come from?
* 289: “all tools search over the same DSL” - have you extended NEAR to use the extra DSL elements, or limited dPads to use NEAR’s DSL?
* 307: Why is the F1 score of 0.475 different from the value in Table 1, 0.458? Is it because Figure 6 contains the best program over 5 runs, and the other runs produce programs with much lower F1 score? What are those other programs?
* Table 2: Is there a copy/paste error here? The F1, Acc, and Time columns are exactly the same between Crim13 and Sk152.
* What are the main differences between dPads and NEAR? Intuitively speaking, why are dPads’ approach better than NEAR’s approach? I understand the argument of being more efficient in an enormous search space, but why is NEAR faster than dPads on Fly-vs-fly, and similar in speed on Crim13 and Sk152? From the timing results, I infer that NEAR must have other kinds of optimizations, and a deeper dive into the pros and cons of each approach is needed.
* Each dataset has a training, validation, and test set. Furthermore, the training set is split into $D_{val}$ and $D_{train}$ (line 279), so now we have 4 splits. Where is each split used?

**Minor suggestions**
* 21: achieves -> achieving
* 32: inefficient in natural -> inefficient in nature? Or just inefficient
* 138: still not sure what “inefficient in natural” means
* 223: depcited -> depicted
* I would give an example task in the introduction or problem formulation to help clarify what is meant by “sequence classification,” which I initially interpreted to mean classification of text. This also helps the reader think about the correct datatypes when reading about the approach
* I would make a connection between iterative graph unfolding and beam search -- when I realized the similarities, the iterative graph unfolding step suddenly made much more sense.


**Time Spent Reviewing:**

6

---

> ### Author Response · Authors · 2021-08-10
> **Review Response**
>
> We appreciate your effort in providing detailed and helpful reviews. We address the concerns and questions as follows:
>
> ## Q1: How does dPads differ from NEAR? Why does dPads outperform NEAR?
>
> Although dPads and NEAR both formalize program synthesis as a graph search problem, the two techniques are completely different. We compare dPads and NEAR as follows.
>
> **NEAR**. Typically, search-based program synthesizers enumerate the underlying program space in some order and for each program checks whether or not it satisfies the synthesis constraints. It is a challenging problem because the architecture search space is combinatorial. The most simple strategy that starts by searching for programs with 1 DSL production rule and iteratively increases this bound does not scale to complex programs. NEAR uses neural models to approximate unexpanded subexpressions to estimate the likelihood of eventually deriving a high-quality program by choosing a particular production rule. It leverages this kind of information to prioritize promising search directions and hence can greatly accelerate the search process. However, NEAR's search strategy is still based on enumeration. When the search space is intractably large, enumeration-based strategies are inefficient in general.
>
> **dPads's Contributions**. dPads proposes a new, scalable program synthesis technique. It views program architecture search as learning a probability distribution over *all* possible program architectures induced by a DSL. Unlike NEAR that enumerates and evaluates each program by training unknown parameters from scratch, dPads reduces the computation cost by training one program, a.k.a. program derivation graph, to approximate the performance of every program in the search space. It learns the architecture weights of each program encoded in a program derivation graph in a way that ranks the performance estimation of these programs. The search procedure in dPads is more efficient because it supports gradient-based architecture optimization in a novel continuous relaxation of the program architecture search space.
>
> **dPads's Impacts.** To the best of our knowledge, dPads is the *first* program synthesis technique that applies gradient-based architecture search. Experiment results on sequence classification demonstrate that thanks to this strategy, dPads can efficiently search in a much deeper program space to learn more complex programs that are necessary to ensure higher F1 scores. NEAR often cannot reach the necessarily deep program space within the time budget because of its less efficient enumeration-based search procedure, resulting in suboptimal programs (see Q2 and Table 8 in the appendix).
>
> ## Q2: If dPads outperforms NEAR, why is NEAR faster than dPads on Fly-vs-fly, and similar in speed on Crim13 and Sk152?
>
> In our experiments, compared to NEAR, dPads can more efficiently reach a much deeper program space (measured by abstract syntax tree heights). The results are as follows (also given in Table 8 of the appendix) where we show for each synthesized program its accuracy, F1 score, the averaged number of DSL production rules used to derive the program (#R), and the running time (#T). Costs of time are in minutes.
>
> |             | Crim13 |      |         |         | Fly-vs-fly |      |         |         | Bball |      |        |         | SK152 |      |         |         |
> |-------------|--------|------|---------|---------|------------|------|---------|---------|-------|------|---------|---------|-------|------|---------|---------|
> |             | **F1**     | **Acc.** | **#R** | **#T** | **F1**         | **Acc.** | **#R** | **#T** | **F1**    | **Acc.** | **#R** | **#T** | **F1**    | **Acc.** | **#R** | **#T** |
> | **A\*-NEAR**     | .286   | .820 | 6.8     | 164.92 | .828       | .764 | 2.8     | 243.82 | .940  | .934 | 8.0     | 553.01 | .312  | .315 | 4.2     | 210.23 |
> | **IDS-BB-NEAR** | .323   | .834 | 5.8     | 463.36 | .822       | .750 | 2.4     | 465.57 | .793  | .768 | 7.0     | 513.33 | .314  | .317 | 4.4     | 848.44 |
> | **dPads**       | .458   | .812 | 10.2    | 147.87 | .887       | .853 | 6.8     | 348.25  | .945  | .939 | 8.0     | 174.68 | .337  | .337 | 7.6     | 162.70 |
>
> On Fly-vs-fly, although NEAR runs faster, it only finds programs derived by ~2.8 production rules but dPads finds much deeper programs derived by ~6.8 production rules. Consequently, for this benchmark, dPads achieves much higher accuracy and F1 scores. A similar trend can be observed for the results on Crim13 and Sk152. On Basketball, both tools find programs with ~8 production rules. In this case, the architecture spaces searched by the two tools are roughly equivalent, but dPads is 3 times faster.
>
> The experiment results consistently demonstrate that dPads's gradient-based architecture search is much more efficient than NEAR's enumeration-based strategy. Moreover, NEAR uses neural models to estimate the performance of *a* partially expanded program. Our experiments find that due to overfitting or underfitting, such a neural model may be biased on *a* particular program. For example, on Fly-vs-fly, due to the biased estimation, NEAR stops searching the architecture space deeper than that contains programs derived by only ~2.8 production rules. In contrast, (1) dPads uses a sub program derivation graph to estimate the performance of a partially expanded program that can provide more accurate assessment due to the graph's syntax resemblance to a valid program; (2) dPads only uses neural models to provide "contrastive" performance estimation for a *set* of programs sharing nodes in a program derivation graph when iteratively unfolding the graph. As a result, on Fly-vs-fly, dPads searches the architecture space much deeper containing programs derived by ~6.8 production rules. Even dPads typically searches much deeper, it often runs faster than NEAR.
>
> In the sequence classification tasks, although dPads learns more complex programs, synthesized programs' classification error plus architecture cost (i.e. the search objective) is still lower than that of the programs learned by NEAR. This demonstrates that dPads synthesizes more complex programs that are necessary to ensure higher F1 scores.
>
>
> ## Q3: Does NEAR already adopt any similar strategies? How novel are dPads's search strategies?
>
> **NEAR does not attempt any search techniques proposed in dPads.** The NEAR authors acknowledge in their paper (page 9) that they only consider enumeration-based search because DARTS-style, gradient-based architecture search cannot be naturally applied to program synthesis due to the complexity of programming languages (e.g. the DSL in Figure 1). They gave two reasons to justify this claim. Firstly, different sets of operations take different input and output types and may only be available at different points of a program. Secondly, there is no fixed bound on the number of expressions in a program architecture.
>
> **dPads's search strategy is novel** because it overcomes the two challenges raised in the NEAR paper. For the first time, dPads demonstrates that gradient-based architecture search is applicable to program synthesis and it substantially outperforms enumeration-based program learning methods including NEAR (reasons are explained in Q1). Node sharing and iterative graph unfolding are two optimizations that further scale dPads to complex programs. Observe that the size of a program derivation graph grows exponentially large with the height of its search tree and the number of functions in a DSL. If without the two optimizations, limited by the size of GPU memory, dPads cannot search programs with deep structures. These two techniques do not appear in NEAR as they are specific to dPads's gradient-based architecture search.
>
> ## Q4: Is dPads's A$^*$ Search any different than NEAR's search?
>
> It appears that dPads and NEAR both use A$^*$ with admissible heuristics to perform a search within a program derivation graph. However, the purposes are quite different. dPads uses A$^*$ to overcome one potential disadvantage of differentiable architecture search: the performance estimation ranked by architecture weights in a program derivation graph can be inaccurate due to the co-adaption among operations (node sharing). While one super program derivation graph may not be able to model the entire search space accurately, dPads uses multiple sub program derivation graphs to effectively address the limitation by having each sub graph modeling one part of the search space. As a node in a program derivation graph contains multiple operations (e.g. Figure 3), dPads separates the entire search space into disjoint partitions by picking one operation from the compound node and assigns a sub program derivation graph to model each partition. This procedure is done in a recursive manner using A$^*$ search to prioritize partitions that include optimal programs.
>
> Moreover, unlike NEAR, the A$^*$ search in dPads does not use neural models to estimate the performance of a partially expanded program. This is because our experiments find that due to overfitting or underfitting, a neural model may be biased on a particular program. In contrast, dPads uses a sub program derivation graph to estimate the performance of a partially expanded program. We find that it can provide more accurate assessment due to the graph's syntax resemblance to a valid program. We only use neural models to provide "contrastive" performance estimation for a *set* of programs sharing nodes in a program derivation graph for iterative graph unfolding (see Q2 for more explanation using the Fly-vs-fly benchmark). We will provide additional details in the final version.

---

> > ### Author Response · Authors · 2021-08-10
> > **Review Response (Continue)**
> >
> > ## Q5: The paper needs a significant change to present the approach in a way that adequately explains its relationship to NEAR.
> >
> > We thank the reviewer for the thoughtful comment. We disagree that the paper needs a major update to illustrate the relationship to NEAR. The two methods are substantially different. As explained in Q1 and Q2, NEAR enumerates the underlying program space in some order and for each program checks whether or not it satisfies the synthesis constraints by training unknown parameters from scratch. In contrast, dPads reduces the computation cost by training one program, a.k.a. program derivation graph, to approximate the quality (e.g. classification accuracy) of every program in the search space.
> >
> > In the paper, Section 1 summarizes the aforementioned key differences, points out that enumeration-based strategies are inefficient in general since the search space can be intractably large, motives how and why gradient-based architecture search in dPads can come to rescue. In Section 3 (the main technical section), we focus on dPads's optimization algorithms that scale gradient-based architecture search to complex program synthesis problems. This section additionally describes how dPads uses graph partitioning (guided by A$^*$ search) to address the shortcoming of only using one program derivation graph to model the entire search space. Since these search techniques are specific to dPads, we do not need to discuss NEAR in the main body of the paper in depth.
> >
> > To address the reviewer's concern, we will give a more detailed review of NEAR in Section 1 to help readers more deeply understand the challenges faced by enumeration-based program architecture search methods including NEAR. We will leverage this detailed review to further help readers obtain stronger intuitive understandings of how and why dPads's search strategies overcome the challenges. We hope this would meet the reviewer's expectations.
> >
> > ## Q6: Each dataset has a training, validation, and test set. Furthermore, the training set is split into $D_{val}$ and $D_{train}$. Where is each split used?
> >
> > We give the full details of each split in line 589 of the appendix. In our experiments, we use the training datasets to optimize the architecture weights and unknown program parameters in a program derivation graph. During training, we additionally split the training set into $D_{val}$ and $D_{train}$ for optimizing architecture weights and unknown program parameters respectively (Equation 2). To avoid overfitting, when learning final programmatic classifiers during A$^*$ search, we use the validation datasets to obtain F1 scores to calculate the A$^*$ heuristic function $h$ (line 239 of the paper) on an optimized sub program derivation graph. We use the test datasets to obtain the final accuracy and F1 scores of synthesized programmatic classifiers.
> >
> > ## Q7: At line 307, why is the F1 score 0.475 of a program for Crim13 different from the value in Table 1, 0.458? Is it because Figure 6 contains the best program over 5 runs, and the other runs produce programs with much lower F1 score?
> >
> > In Table 1, we report the F1 score 0.458 as the average of 5 random runs. In Figure 6, we give one program learned for Crim13 that has an F1 score 0.475. We show two other programs learned for Crim 13 on page 19 of the appendix. All the 5 programs synthesized by dPads have similar F1 scores 0.4676, 0.4752, 0.4558, 0.4427, 0.4471 as depicted in Figure 5(a). Figure 5 demonstrates that dPads consistently outperforms NEAR in achieving higher F1 scores and has significantly lower variance in F1 scores.
> >
> > ## Q8: Where do the costs $c(r)$ come from?
> >
> > We explain in line 622 of the appendix our definition of the costs $c(r)$.  We set $c(r) = 1$ for any grammar rule $r$ and aim to learn a program that can be derived by as few grammar rules as possible. We report the average numbers of grammar rules used to construct learned programmatic classifiers in Table 8 of the appendix (See Q2 for more details).
> >
> > ## Q9: Have you extended NEAR to use the extra DSL elements, or limited dPads to use NEAR’s DSL?
> >
> > For a fair comparison, we limit dPads to use exactly the same DSLs used to evaluate NEAR.
> >
> > ## Q10: Table 2: The F1, Acc, and Time columns are exactly the same between Crim13 and Sk152.
> >
> > We apology for the confusion. We have corrected Table 2 in Appendix A.
> >
> > **Other Comments.** In addition to these main points, we want to thank the reviewer for the helpful suggestions on how to further improve the paper. We plan on incorporating all of these, such as more thoroughly comparing dPads to NEAR early in the paper, making a connection between iterative graph unfolding and beam search, and pointing out that dPads requires more time and resources to produce a program compared to plain RNN methods.

---

> ### Author Response · Authors · 2021-08-25
> **Looking forward to further discussions!**
>
> Dear Reviewer,
>
> We thank you for your constructive comments again. We summarize some of your major concerns and our responses as follows.
>
> The reviewer's first major concern is the similarity between dPads and NEAR. We tried to address this concern in Q1 and Q3 in the previous author response. Notably, the NEAR authors explicitly acknowledge in their paper (page 9) that they do *not* consider DARTS-style, gradient-based architecture search and instead use enumeration-based search because they deem DARTS-style search cannot be naturally applied to program synthesis due to the complexity of programming languages. However, dPads overcomes this challenge and, for the first time, demonstrates that gradient-based architecture search is applicable to program synthesis. This argument strongly supports our claim of the novelty of dPads and the significant difference between dPads and NEAR. Our contributions also include node sharing and iterative graph unfolding, two unique optimizations (not seen in NEAR) that further scale dPads to complex programs. They are specific to dPads and cannot be simply applied to NEAR.
>
> The reviewer's second major concern is our experimental analysis of the differences between dPads and NEAR. We tried to address this concern in Q2 in the previous author response. In brief, the experiment results in Q2 (data also presented in the paper) confirm that dPads searches deeper than NEAR in the program architecture space to synthesize more sophisticated programs that are necessary to ensure higher F1 scores. On Fly-vs-fly, NEAR's inaccurate admissible neural heuristics prevent it from searching the architecture space deeper than that contains programs derived by just ~2.8 production rules. In contrast, dPads uses a sub program derivation graph to estimate the performance of a partially expanded program. The sub program derivation graph provides a more accurate assessment due to the graph's syntax resemblance to a valid program. As a result, dPads searches the architecture space much deeper containing programs derived by ~6.8 production rules and gets a much higher F1 score (.887 vs .828). Further analysis reveals that when the architecture spaces searched by the two tools are roughly equivalent, dPads is significantly faster (e.g. 3 times faster on Basketball). This is because of dPads's more efficient gradient-based search strategy and its uses of neural networks to provide "contrastive" performance estimation for a set of programs sharing nodes in a program derivation graph (as opposed to a single program in NEAR). In iterative graph unfolding, the latter strategy effectively prunes away a large set of low-quality programs.
>
> We sincerely hope that the reviewer can re-evaluate our paper after reading our responses. We will make the above discussions more explicitly in our paper and incorporate all your suggestions in the final version. More discussions and suggestions on further improving the paper will be appreciated!

---

### Official Review · Reviewer_3c6i · 2021-07-16

**Rating:** 6
**Confidence:** 3

**Summary:**

This paper considers the task of synthesizing differentiable programs for sequence classification tasks. It introduces a novel method, dPads, that first constructs a program derivation graph representing a subset of the highest probability possible architectures under a CFG, and subsequently uses an A*-like search to find a program in the graph that balances minimizing complexity with maximizing accuracy. Central to the derivation graph construction method are the ideas of (1) iterative training of architecture weights and neural approximations of subexpressions, (2) node sharing, and (3) iterative graph unfolding. The method is evaluated on four sequence classification tasks, where it is found to have better performance than the program synthesis baselines considered, and worse performance than an RNN baseline, with the key advantage over the RNN being the interpretability of the resultant programs from dPads.

**Limitations And Societal Impact:**

The authors acknowledge the main technical limitation of the work, the relative performance compared with RNN methods. The authors also mention in Section 6 that there is potential for attacks on security and fairness on models derived via the paper's method.

**Main Review:**

Program synthesis is an important area because it yields results that are more interpretable, manipulable, and potentially generalize more robustly than neural models. So, it is promising that this method produces results beginning to approach those obtained by an RNN on sequence classification tasks. Still, the results are far enough behind the RNN's to make the accuracy / interpretability trade-off significant.

The approach builds upon the ideas of NEAR [1] and Euphony [2]. It uses a similar strategy to NEAR of (1) using neural models to approximate unexpanded subexpressions and (2) using an A*-like search for the optimal program. The contributions of this paper over those of [1] and [2] come from the methods of making the search space smaller. Performing iterative graph unfolding retaining only the top-N programs per node, combined with node sharing, allows for this. The other core aspect of this paper's method is it constructs a continuous relaxation of the otherwise discrete search space, iteratively training the architecture weight selection parameters and the neural subexpression parameters; this is most similar to the approach of the DARTS line of work [3], with which I acknowledge limited familiarity.

To demonstrate the value of this reduced search space, the experiments compare against two methods from [1], making the claim that other methods perform even worse and need not be compared against in the paper. Overall, the evaluation is done well, on four datasets with sequence classification tasks. These tasks are quite similar to one another in number of examples, frames, and number of features, and also share the similarity that they all come from visual recordings over time, despite the approach being more general than this one modality. It is interesting to note that the variance in run time can be quite high. Consider providing an explanation of this observation.

Consider also making the description of T clearer. Line 240 states the intention of T, but making it precise or explicitly stating its type or outputs would improve clarity. Other than this, the paper is written precisely. The appendix provides adequate details to understand the experiments fully.

Consider also including empirical information about the reduction in search space provided by node sharing and top-N retention. While the run times are informative, quantifying the search space reduction would be more useful for understanding the source of the relative performance of dPads and baselines more precisely.

From the results in Appendix A, it is interesting that performance already begins to drop off when N=3. The increase in time is understandable, but the drop in f1 score and accuracy is surprising. Do you have an explanation for this? Much of Appendix B already appears in the paper and is duplicated precisely; this is not necessary.

---

[1] Learning Differentiable Programs with Admissible Neural Heuristics https://arxiv.org/abs/2007.12101

[2] Accelerating Search-Based Program Synthesis using Learned Probabilistic Models https://www.seas.upenn.edu/~mhnaik/papers/pldi18b.pdf

[3] DARTS: Differentiable Architecture Search https://arxiv.org/abs/1806.09055

---

Line 21: s/achieves/achieving

Line 204: "noterminals" should be "nonterminals".

Line 223: "depcited" should be "depicted"

Line 234: $q[u/f^u_k]$ denotes the graph with $u$ modified, not the $f$-score; reword to make this clear.

Line 535: s/shows/show

Line 553: I'm not familiar with the term "sortness".

**Time Spent Reviewing:**

7

---

> ### Author Response · Authors · 2021-08-10
> **Review Response**
>
> We appreciate your effort in providing detailed and helpful reviews. We address the concerns and questions as follows:
>
> ## Q1: dPads's results are far behind the RNN's to make the accuracy / interpretability trade-off significant.
>
> We agree with the reviewer that RNN models in general have better performance than programmatic models learned by dPads. However, in our experiment, we found that on several domains the programmatic models have comparable performance with the RNN models.
> * On the Crim13 domain, the programmatic model in Figure 6(a) achieves a high F1 score 0.475, only 0.006 less than the RNN model. Another synthesized programmatic model on this domain (line 720 in the appendix) achieves a comparable high F1 score 0.468.
>
> * On the Fly-vs-fly domain, the learned programmatic model (line 729 in the appendix) achieves a high F1 score 0.904, only 0.06 less than the RNN model.
>
> * On the Basketball domain, the learned programmatic model (line 747 in the appendix) also achieves a high F1 score 0.945, only 0.035 less than the RNN model.
>
> In general, our results verify that our programmatic models on the above three domains are comparable with highly expressive RNNs. Observe that there is at most an 8\% drop in F1 scores when using the learned programmatic models. Thus, we believe that our results are sufficient to support our claim that dPads discovers programmatic classifiers that yield natural interpretations and achieve competitive quality.
>
> More importantly, dPads's performance outperforms NEAR, the current state-of-the-art method for program learning from a single dataset. dPads opens up a new promising line of research to address the accuracy / interpretability trade-off in Machine Learning.
>
> ## Q2: The benchmarks are quite similar to one another and all come from visual recordings over time.
>
> As the reviewer has pointed out that our approach is more general than this one modality. An important motivation of our work is to synthesize interpretable programs. We focused on behavior analysis applications where interpretability is an important concern. Our datasets are representative of real behavior data used by real domain experts (e.g. sports analysts for basketball, and neuroscientists for Fly-vs-Fly). We will add a sequence regression task to the final version of our paper.
>
> ## Q3: The variance in run time can be quite high.
>
> The variance in run time is high because dPads may synthesize programs with different sizes due to randomness in stochastic gradient descent. For example, we show two programs synthesized by dPads as follows (also available in the appendix line 704, 720).
>
> **Program 1:**
> ```
> Map(
>     if AccelerationSelect$_{\theta_1}$($x_t$)
>     then if PositionSelect$_{\theta_2}$($x_t$)
>          then PositionSelect$_{\theta_3}$($x_t$)
>          else VelocitySelect$_{\theta_4}$($x_t$)
>     else Multiply(DistanceSelect$_{\theta_5}$($x_t$), DistanceSelect$_{\theta_6}$($x_t$))) $x$
>
> ```
> **Program 2:**
> ```
> MapPrefixes(
>     Fold(
>         if DistanceSelect$_{\theta_1}$($x_t$)
>         then PositionSelect$_{\theta_2}$($x_t$)
>         else PositionSelect$_{\theta_3}$($x_t$))) $x$
> ```
> The two programs have similar F1 scores. As a result, during search, the two sub program derivation graphs led by "Map" and "MapPrefixes" also have very similar F1 scores. Due to the randomness in stochastic gradient descent, either one might be a little higher than the other and hence chosen firstly by the search algorithm. dPads spends more time on synthesizing Program 1 because it searches deeper in the program architecture space for this program as the program's abstract syntax tree is deeper. We will provide additional details in the final version.
>
> ## Q4: Empirical information about the reduction in search space provided by node sharing and top-N preservation.
>
> We thank the reviewer for the great suggestion. We quantify the program derivation graph space reduction by node sharing and iterative unfolding as follows. We show for each benchmark the number of nodes (# Node), the total number of DSL functions hosted by the nodes (# Oper.), the total number of edges (# Edges), the number of architecture weights (# Weights), and the number of unknown program parameters (# Param.) to train on its program derivation graph. We only show the results of the deepest program derivation graph generated by dPads on each benchmark.
>
> |                            | Crim13 |         |         |           |                     |  Sk152 |         |         |           |          | Fly-vs-fly |         |         |           |          | Basketball |         |         |           |          |
> |----------------------------|:------:|---------|---------|-----------|---------------------|:------:|---------|---------|-----------|----------|:----------:|---------|---------|-----------|----------|:----------:|---------|---------|-----------|----------|
> |                            | # Nodes | # Oper. | # Edges | # Weights | # Param.            | # Nodes | # Oper. | # Edges | # Weights | # Param. | # Nodes     | # Oper. | # Edges | # Weights | # Param. | # Nodes     | # Oper. | # Edges | # Weights | # Param. |
> | dPads w/o graph unfolding            | 33     | 297     | 32      | 707       | 914                 | 33     | 337     | 32      | 681       | 48636    | 33         | 485     | 32      | 1021      | 12803    | 33         | 248     | 32      | 510       | 6281     |
> | dPads w/o node sharing | 82     | 519     | 81      | 518       | 1814                | 60     | 196     | 59      | 195       | 34252     | 86         | 441     | 85      | 440       | 10612    | 94         | 302     | 93      | 301       | 8336     |
> | dPads            | 15     | 100     | 14      | 203       | 315                 | 15     | 57      | 14      | 89        | 7298     | 21         | 220     | 20      | 370       | 5127     | 29         | 152     | 28      | 259       | 3898     |
>
> Notice that dPads without iterative graph unfolding and dPads without node sharing generate program derivation graphs that are significantly larger.
>
> For example, on Fly-vs-fly, without iterative graph unfolding, the program derivation graph has 33 nodes shared by 485 operations, 32 edges, and a total of 13824 parameters to train. Directly applying dPads to train under this setting encounters an out-of-memory (OOM) exception. In contrast, with iterative graph unfolding, the deepest graph generated by dPads has only 21 nodes shared by 220 operations, 20 edges, and a total of 5497 parameters to train. dPads converges in less than 360 mins.
>
> On Basketball, without node sharing, the deepest program derivation graph generated by dPads has 94 nodes hosting 302 operations, 93 edges, and a total of 8637 parameters to train. If directly running dPads under this setting, the search procedure (Sec 3.3) does not terminate in 24 hours because of the many nodes and edges. In contrast, with node sharing, the program derivation graph reduces to only 29 nodes shared by 152 operations, 28 edges, and a total of 4157 parameters. dPads converges in less than 180 mins.
>
> We will add the quantification results into the final version of the paper.
>
>
> ## Q5: F1 score and accuracy begin to drop off when N=3.
>
> The result of our ablation study is duplicated as follows to ease communication. Our interpretation of the result is that dPads gets similar accuracy and F1 scores when setting N = 3 compared to N = 2 but consumes more time.
>
> |        | N | dPads |      |        |         |           |
> |:------:|---|-------|------|--------|---------|-----------|
> |        |   | **F1**    | **Acc.** | **Time**   | **Std. F1** | **Std. Acc.** |
> | **Crim13** | 1 | .272  | .627 | 50.85  | .111    | .218      |
> |        | 2 | .458  | .812 | 147.87 | .014    | .008      |
> |        | 3 | .450  | .811 | 441.12 | .025    | .008      |
> | **Sk152**  | 1 | .310  | .310 | 40.34  | .020    | .024      |
> |        | 2 | .337  | .337 | 162.70 | .017    | .017      |
> |        | 3 | .336  | .338 | 609.14 | .011    | .010      |
>
>
> We appreciate it if the reviewer could provide some clarification of the question in the rolling discussion phase as we do not observe that F1 score or accuracy begins to drop off when N = 3.
>
> ## Q6: Line 240: The description of $\mathcal{T}$ is not very clear
>
> $\mathcal{T}$ is encoded exactly using Equation (3) in line 178 of the paper. We will provide additional details in the final version.
>
> ## Q7: Much of Appendix B appears in the paper and is duplicated.
>
> In Appendix B, we have provided additional theoretical analysis on why dPads can optimally balance structure cost and program performance.
>
> **Other Comments.** In addition to these main points, we want to thank the reviewer for the helpful suggestions on how to further improve the paper. We plan on incorporating all of these, such as explaining the high variance in run time (Q3) and including empirical information about the reduction in search space provided by node sharing and top-N preservation (Q4). We will also clarify the accuracy / interpretability trade-off (Q1).

---

> > ### Comment · Reviewer_3c6i · 2021-08-24
> > **Thanks for your review response**
> >
> > Thanks for your review response.
> >
> > I appreciate in particular your discussions of the accuracy/interpretability tradeoff, the set of tasks considered, and the reasons for variance in dPads runtimes. The empirical search space data is also much appreciated.
> >
> > The authors write in their response:
> >
> > > dPads opens up a new promising line of research to address the accuracy / interpretability trade-off in Machine Learning.
> >
> > I agree, this is indeed a promising direction for research.
> >
> > Regarding "Q5", I agree with your assessment of the results. Between Crim13 and Sk152, three of four of the reported f1/accuracy metrics decrease from N=2 to N=3 (in the original Table 2, with the copy/paste error, this was four of four), but they do so by values well within the standard deviations of these metrics.

---

> > > ### Author Response · Authors · 2021-08-25
> > > **We appreciate your comments**
> > >
> > > We apology for the copy/paste error. Many thanks for your constructive comments! We will incorporate all of your suggestions into our paper.

---

### Official Review · Reviewer_twpr · 2021-07-17

**Rating:** 7
**Confidence:** 3

**Summary:**

This work proposes a method for differentiable synthesis of program architectures. A program derivation graph is encoded as a differentiable program whose output is a weighted output of all the programs in the graph. The method learns a probability distribution over all possible program derivations induced by a context-free grammar. The grammar rule selection weights and the program parameters are co-optimized iteratively with respect to a loss function over program outputs. The approach is evaluated on four sequence classification tasks. The discovered architectures have better F1 scores as compared to the state-of-the-art program synthesis methods (A*-NEAR and IDS-BB-NEAR).


**Limitations And Societal Impact:**

* The paper doesn’t spend much time discussing the limitations or possible pain points. This is, in one way, a byproduct of the inadequate evaluations. The proposed approach is only compared with NEAR. I believe more diverse tasks of varying complexity will reveal the strong and weak points of the method.
* The paper already acknowledges that the results achieved with dPads do not match the RNN baseline result. This may be due to limitations in expressivity imposed by the DSL. I would like the authors to comment on this.

There does not seem to be any potential negative societal impact of this work.


**Main Review:**

**Originality**

The idea of differentiable network architecture search by relaxing the search space to be continuous has already been explored in DART. However, applying this technique to architecture search is not straightforward; the space of program architectures in this setting is huge. This work introduces two techniques, "node sharing" and "iterative unfolding" to address the challenges associated with program synthesis of differentiable programs. If the expansion of nonterminals in two programs are expanded with the same grammar production rule then the resulting subterms can be clubbed together in one node. This converts the program expansion tree into a graph reducing the search space to a great extent.
The Iterative Graph Unfolding approach unfolds the program derivation graph iteratively and prunes away unlikely candidates architectures at the end of each iteration. As the rule selection probabilities and program parameters are co-optimized, iterative graph unfolding ensures that only relevant nodes are kept in the program derivation tree. In my view, these two techniques are the main contributions of this work.

**Significance**

Program Architecture Search is a challenging problem due to the exponential search space. The enumeration-based strategies just do scale to the real world tasks. Differentiable synthesis is an important research direction for effective search. The two techniques introduced in the papers, "node sharing" and "iterative graph unfolding" enables the use of differentiable search methods for synthesizing programmatic models from a context free grammar.

**Quality**

* The problem formulation and the method presentation is technically sound. The iterative parameter update rules are presented clearly.
* The proposed method has been implemented as a tool called dPads. dPads is evaluated on four sequence classification tasks. The proposed method outperforms the state-of-the-art program synthesis methods on the F1 score metric and time.
* I think the evaluation is a bit inadequate. Although the datasets are from diverse domains, I would have liked some variety in the tasks. Currently all the tasks are sequence classification tasks. (Maybe include regression and other data types in addition to sequences). Some comparison with the enumerative approaches would have emphasized the benefit of the proposed approach.
* Naive application of DART like differentiable search does not work in the program synthesis domain. Including the results on dPads without using node sharing and/or iterative folding would bolster the paper's claim.
* The authors might consider reporting ROC-AUC to report the results across thresholds (if the dataset is not imbalanced.)

**Clarity**
* The paper is well written and easy to follow. I would have liked some light introduction to the Program Derivation Graph before diving into the notation heavy exposition. You may think of presenting a simple program derivation tree and the corresponding graph with collapsed nodes.
* I do not get the sentence on line 200. Is it that each node has at most N subterms?  However, there are 3 subterms in node 4 in Figure 4 (step 5).
* Minor: Line 85 uses $\sigma$ for grammar vocabulary. Immediately after that $\sigma$ is used for sigmoid function.
* Minor: Figure 5. Swap figures b and c to keep the same sequence as in Table 1.


**Time Spent Reviewing:**

8

---

> ### Author Response · Authors · 2021-08-10
> **Review Response**
>
> We appreciate your effort in providing detailed and helpful reviews. We address the concerns and questions as follows:
>
> ## Q1: Currently all the tasks are sequence classification tasks.
>
> An important motivation of our work is to synthesize interpretable programs. We focused on behavior analysis applications where interpretability is an important concern. Our datasets are representative of real behavior data used by real domain experts (e.g. sports analysts for basketball). We will add a sequence regression task to the final version of our paper.
>
> ## Q2: No comparison with any enumerative approaches.
>
> We have compared dPads with an enumeration strategy that synthesizes and evaluates complete programs in order of increasing complexity. This strategy is widely used in program synthesis tasks. We set the running time of the enumeration strategy twice as long as dPads's synthesis time.
>
> |                  | Crim13 |      | Fly-vs-fly |  | Bball | | Sk152 |   |
> |:----------------|:------:|:----:|:-------------:|:----------:|:----:|:------:|:-----:|:----:|
> |                  | **F1**     | **Acc.** | **F1** | **Acc.** | **F1**    | **Acc.** | **F1**    | **Acc.** |
> | **Enumeration** | .294 | .856 | .850 | .774 | .795  | .767 | .288  | .284 |
> | **dPads**            | .458   | .812 | .887 | .853 | .945  | .939 | .337  | .337 |
>
> dPads outperforms the enumeration strategy on all of the four benchmarks. Although the enumeration strategy gets higher accuracy on Crim13, it underfits this unbalanced dataset as the F1 score is very low. We have also tried a Monte-Carlo tree search strategy. Surprisingly, its performance is even worse than the simple enumeration strategy. We will provide additional details in the final version.
>
> ## Q3: No ablation study on dPads without node sharing and/or iterative unfolding.
>
> We started our research without considering node sharing and iterative graph unfolding but found that dPads does not work well without these two optimization strategies. The reason is that the size of a program derivation graph grows exponentially large with the height of its search tree (e.g. Figure 2) and the number of DSL functions (e.g. Fold and ITE). Without the two optimizations, limited by the size of GPU memory, dPads may either time-out or encounter out-of-memory error when searching programs that need deep structures to ensure high accuracy. We report the comparison results over 5 random runs as follows. Costs of time are set in minutes.
>
> |                  | Crim13 |      |        | Fly-vs-fly |      |        | Bball |      |        | Sk152 |      |        |
> |:----------------|:------:|:----:|:------:|:----------:|:----:|:------:|:-----:|:----:|:------:|:-----:|:----:|--------|
> |                  | **F1**     | **Acc.** | **Time**   | **F1**         | **Acc.** | **Time**   | **F1**    | **Acc.** | **Time**   | **F1**    | **Acc.** | **Time**   |
> | **dPads w/o node sharing**  | .453   | .800 | 334.93 | -          | -    | >24hrs | -     | -    | >24hrs | .321  | .322 | 252.81 |
> | **dPads w/o graph unfolding** | .449   | .818 | 280.67 | -          | -    | OOM    | .848  | .832 | 348.09 | .348  | .346 | 273.95 |
> | **dPads**            | .458   | .812 | 147.87 | .887       | .853 | 348.25 | .945  | .939 | 174.68 | .337  | .337 | 162.70 |
>
>
> The results demonstrate that dPads with the two optimization strategies converges much faster.
>
> On Fly-vs-fly, without iterative graph unfolding, the program derivation graph generated by dPads has 33 nodes shared by 485 operations, 32 edges, and a total of 13824 parameters to train. dPads encounters an out-of-memory (OOM) exception when training this graph. In contrast, with iterative graph unfolding, the deepest graph has only 21 nodes shared by 220 operations, 20 edges, and a total of 5497 parameters to train. dPads converges in less than 360 mins.
>
> On Basketball, without node sharing, the deepest program derivation graph generated by dPads has 94 nodes hosting 302 operations, 93 edges, and a total of 8637 parameters to train. The search procedure of dPads (Sec 3.3) does not terminate in 24 hours because of the many nodes and edges. In contrast, with node sharing, the program derivation graph reduces to only 29 nodes shared by 152 operations, 28 edges, and a total of 4157 parameters. dPads converges in less than 180 mins.
>
> Moreover, training without these two optimizations does not necessarily produce better results even when there is not OOM or timeout. For example, on Basketball, dPads achieves .945 F1 score. dPads without iterative graph unfolding only obtains .848 F1 score. We suspect this is because the program derivation graph without iterative unfolding is more difficult to train as it contains significantly more parameters (6791 vs 4157). In contrast, dPads uses top-N preservation to efficiently prune away low-quality options. We will provide additional details in the final version.
>
> ## Q4: Lack of discussion on the limitations of dPads.
>
> **Performance gap to RNN.** As the reviewer has pointed out, our paper acknowledged that dPads's performance does not match the RNN baseline. This is mainly due to the limitations in expressivity imposed by the DSL. Firstly, in the DSL, we only allow for customized feature functions $F_{S,\theta}(x)$ that extract a vector consisting of a predefined subset $S$ of the dimensions of an input sequence $x$ and pass the extracted vector through a linear function with trainable parameters $\theta$. For example, for Crim13, we predefine feature functions such as the XY positions, angles, velocity, acceleration, distance of a pair of mice, and distance difference for every two consecutive frames. We list the details of $F_{S,\theta}(x)$ for each dataset in Table 3, 4, 5, 6 in Appendix C. These feature functions are extremely helpful to ensure that a synthesized program composed of these functions is interpretable. However, a program limited to these predefined feature functions may be suboptimal as only a subset of features is used. Instead, an RNN policy can understand the whole context of a sequence using all features available from the input. Secondly, for the sake of interpretation, our DSL also predefines a limited set of algebraic operations to process the outputs from the feature functions. However, an RNN can use a more expressive activation function to learn about long-term dependencies in data. We have reported the performance limitation of dPads in Table 1.
>
> **Program Derivation Graph Accuracy.** Another important limitation is that performance estimation of each program included in a program derivation graph ranked by architecture weights can be inaccurate due to the co-adaption among operations (node sharing). As one super program derivation graph may not be able to model the entire search space accurately, we have used multiple sub program derivation graphs generated on the fly via a variant of A$^*$ search to address this limitation (Section 3.3). Each sub derivation graph models one part of the search space. However, as reported in the ablation study (Table 2 and Appendix A), the A$^*$ search slows down the whole synthesis procedure, despite improving the accuracy, and may need additional optimization.
>
> ## Q5: Does each node have at most N subterms after top-N preservation?
>
> After top-N preservation, each node may contain more than N subterms. We preserve the top-N subterms in a node $n$ for each term in $n$'s parent. For example, consider node 3 of the program derivation graph in Figure 3. The parent of node 3 is node 1. Assume N=1. After top-1 preservation on node 3, it contains 2 subterms *x* and *Fold* according to the $w$ matrix in Figure 3. *x* is retained on node 3 for the *Add* function on the parent node 1 and *Fold* is retained for the *ITE* function on node 1. We will provide this example in the final version.
>
> **Other Comments.** In addition to these main points, we want to thank the reviewer for the helpful suggestions on how to further improve the paper. We plan on incorporating all of these, such as a light introduction to the Program Derivation Graph including a simple program derivation tree and the corresponding graph, correcting the use of $\sigma$ for various purposes, swapping Figure 5(b) and Figure 5(c) to keep the same sequence as in Table 1. The detailed discussion on further clarifications is deeply appreciated as well. We will add comparison with the enumerative approaches to emphasize dPads's benefits (Q2), include the results without node sharing and/or iterative unfolding (Q3), and discuss the limitations of dPads in depth (Q4) in the final version.

---

> > ### Comment · Reviewer_twpr · 2021-08-26
> > **Thank you for the review response**
> >
> > Thank you for the detailed review response.
> >
> > I appreciate the results on comparison with the enumerative approach and ablation studies.  It is important that you include the limitations in the final version.

---

> > > ### Author Response · Authors · 2021-08-26
> > > **Thank you for your suggestions**
> > >
> > > We will include the results on comparison with the enumerative approach and the ablation studies, as well as the discussion on the limitations in the final version. We appreciate your constructive comments!

---

### Decision · Program_Chairs · 2021-09-27

**Decision:**

Accept (Poster)

**Comment:**

The reviewers appreciated the novel idea presented in the paper to use gradient-based search in a continuous relaxation of context-free grammar rules for synthesis of program architectures. The additional experiments with comparison to enumerative approaches and ablation experiments, as well as more information about empirical savings in search space and relationship with NEAR were also greatly appreciated. There was still a concern about improving the writing and presentation in the paper, which hopefully the authors can improve in the final version taking feedback from the detailed reviews. It would also be great to incorporate the new experiments and discussions from the responses in the final version.